



# Wind, waves, and surface currents in the Southern Ocean: observations from the Antarctic Circumnavigation Expedition

**Marzieh H. Derkani[1], Alberto Alberello[2,3], Filippo Nelli[1], Luke G. Bennetts[3], Katrin G. Hessner[4], Keith MacHutchon[5], Konny Reichert[6], Lotfi Aouf[7], Salman Khan[8], and Alessandro Toffoli[1]**

[1]Department of Infrastructure Engineering, The University of Melbourne, 3010, Melbourne, Victoria, Australia
[2]Department of Physics, University of Turin, 10125, Turin, Italy
[3]School of Mathematical Sciences, University of Adelaide, 5005, Adelaide, South Australia, Australia
[4]OceanWaveS GmbH, 21339 Lüneburg, Germany
[5]Department of Civil Engineering, University of Cape Town, 7701, Cape Town, South Africa
[6]Independent Scholar, 6021 Wellington, New Zealand
[7]Météo-France, 31100, Toulouse, France
[8]Oceans and Atmosphere, Commonwealth Scientific and Industrial Research Organisation, 3195, Aspendale, Victoria, Australia

**Correspondence:** Marzieh H. Derkani (marzieh.h.derkani@gmail.com) and
Alessandro Toffoli (toffoli.alessandro@gmail.com)

**Abstract.** The Southern Ocean has a profound impact on the Earth's climate system. Its strong winds, intense currents, and fierce waves are critical components of the air–sea interface and contribute to absorbing, storing, and releasing heat, moisture, gases, and momentum. Owing to its remoteness and harsh environment, this region is significantly undersampled, hampering the validation of prediction models and large-scale observations from satellite sensors. Here, an unprecedented data set of simultaneous observations of winds, surface currents, and ocean waves is presented, to address the scarcity of in situ observations in the region – https://doi.org/10.26179/5ed0a30aaf764 (Alberello et al., 2020c) and https://doi.org/10.26179/5e9d038c396f2 (Derkani et al., 2020). Records were acquired underway during the Antarctic Circumnavigation Expedition (ACE), which went around the Southern Ocean from December 2016 to March 2017 (Austral summer). Observations were obtained with the wave and surface current monitoring system WaMoS-II, which scanned the ocean surface around the vessel using marine radars. Measurements were assessed for quality control and compared against available satellite observations. The data set is the most extensive and comprehensive collection of observations of surface processes for the Southern Ocean and is intended to underpin improvements of wave prediction models around Antarctica and research of air–sea interaction processes, including gas exchange and dynamics of sea spray aerosol particles. The data set has further potentials to support theoretical and numerical research on lower atmosphere, air–sea interface, and upper-ocean processes.

## 1   Introduction

The Southern Ocean comprises an uninterrupted band of water around Antarctica south of the 60th parallel. More broadly, it refers to the body of water south of the main landmasses of Africa, Australia, and South America, with a northern limit at approximately 40° S (see, for example, Young et al., 2020). This region is dominated by strong westerly winds, the notorious roaring forties, furious fifties, and screaming sixties (Lundy, 2010). They fuel the Antarctic Circumpolar Current (the world's largest ocean current, e.g. Park et al., 2019), which mixes warm waters descending from the Atlantic, Indian, and Pacific oceans with northward cold streams from the Antarctic. Above all, intense winds give rise to some of the fiercest waves on the planet (e.g Barbariol et al., 2019; Vichi et al., 2019; Young and Ribal, 2019; Young et al., 2020). Acting as an interface between the lower atmosphere and the upper ocean, waves entrap and release momentum, heat, moisture, and gases through breaking (Melville, 1996; Csanady, 2001; Veron, 2015) and drive air–sea fluxes (e.g. Humphries et al., 2016; Schmale et al., 2019; Thurnherr et al., 2020). Due to almost unlimited fetches (the distance of open water over which the wind blows), Southern Ocean waves are normally long and fast moving, allowing them to inject turbulent motion throughout the water column down to depths of 100–150 m, i.e. approximately half wavelength, and contributing to ocean mixing (Babanin, 2006; Qiao et al., 2016; Toffoli et al., 2012; Alberello et al., 2019b). The combined effect of the Antarctic Circumpolar Current, which cools and sinks near-surface water, and waves, which regulate fluxes and stir the upper ocean, produces a well-mixed layer that extends from about 100 m in the summer months to approximately 500 m in the winter months (e.g. Dong et al., 2008). This deep mixed layer gives the Southern Ocean capacity to store more heat and gases than any other latitude band on the planet, making this remote ocean a major driver of the Earth's climate system (see, for example, Dong et al., 2007).

South of the 60th parallel, a strong sea ice seasonal cycle of advance and retreat (Eayrs et al., 2019) forms an integral part of a coupled atmosphere-sea-ice–ocean system and influences Southern Ocean dynamics. Sea ice extent around Antarctica impacts albedo, atmospheric and thermohaline circulation, and ocean productivity (Perovich et al., 2008; Massom and Stammerjohn, 2010; Notz, 2012), contributing to the heat balance. Further, it attenuates waves, modulating air–sea fluxes and mixing (Thomas et al., 2019). In turn, waves (in combination with wind) have a significant feedback to the Antarctic sea ice state, extent, and thickness (e.g Wadhams, 1986; Bennetts et al., 2017; Alberello et al., 2019a; Vichi et al., 2019; Alberello et al., 2020a).

In situ observations of atmospheric and oceanographic properties are scarce due to the remoteness of the region. Although there have been many expeditions crossing the Southern Ocean (see a general overview in Schmale et al., 2019), measurements have primarily been taken en route to Antarctic stations, leaving entire sectors undersampled. Further, measurements normally concentrate on the lower atmosphere and/or the upper ocean (not necessarily concomitantly), while waves are generally not monitored. Only a handful of buoys have operated in the region: (i) the Southern Ocean Flux Station (Schulz et al., 2011, 2012), a meteorological buoy first deployed in 2010 at approximately 350 nautical miles south-west of Tasmania (Australia) that provides observations of meteorological parameters, including the directional wave spectrum, downwelling radiation, and seawater temperature and salinity; (ii) the Southern Ocean wave buoy network, which is comprised of one directional wave buoy deployed south of Campbell Island (New Zealand) and five drifting buoys (Barbariol et al., 2019); and (iii) the Global Southern Ocean Array and the Global Argentine Basin Array, which are networks of fixed and moored platforms and mobile profilers (gliders) deployed south-west of Chile and in the Argentinian basin, respectively, to monitor waves, air–sea fluxes of heat, moisture and momentum, and physical, biological, and chemical properties throughout the water column (Trowbridge et al., 2019). Buoys have also been deployed in the Antarctic marginal ice zone to monitor waves in ice and sea ice drift (e.g. Meiners et al., 2016; Ackley et al., 2020; Meylan et al., 2014; Vichi et al., 2019; Alberello et al., 2020a).

A database covering the Southern Ocean more uniformly is provided by polar-orbiting microwave radar satellites such as altimeters, scatterometers, and synthetic aperture radar (SAR). Nevertheless, sea state observations are scattered in both space and time due to the nature of polar satellite orbits, and normally limited to average wind and current speeds, and wave heights. SAR technology provides images that can be converted into directional wave energy spectra (Khan et al., 2020a). However, SAR only detects swell systems, i.e. long-wave systems no longer under the effect of local winds and with wavelengths longer than 115 m (Collard et al., 2009). It does not resolve the wind sea, i.e. the short-wave components directly generated by local winds. This limitation is partly addressed by the recently launched Chinese-French Oceanography Satellite (CFOSAT) mission, which detects wave systems with wavelengths longer than 70 m (Hauser et al., 2020; Aouf et al., 2020).

The scarcity of in situ observations has a negative feedback on the satellite network, which cannot rely on sufficient ground truth to be validated with high confidence. In turn, this drawback impacts prediction models, which are impaired by notable biases in the Southern Ocean (see, for example, Yuan, 2004; Li et al., 2013; Zieger et al., 2015). To address the lack of in situ observations and support calibration and validation of satellite sensors and prediction models, an international initiative, organised by the Swiss Polar Institute, led an unprecedented circumnavigation of the Antarctic continent (the Antarctic Circumnavigation Expedition, ACE; Walton and Thomas, 2018) during the Austral summer of

2016–2017. The objective of the expedition was to sample concomitant processes in the lower atmosphere, at the ocean surface, and in the upper-ocean layers all around the Antarctic continent in a single season (Rodríguez-Ros et al., 2020; Schmale et al., 2019; Smart et al., 2020; Thurnherr et al., 2020). Here we present a database of underway sea state observations, comprising of concurrent records of winds, surface currents, and waves. In Sects. 2 and 3, details of the expedition, the instrumentation, and its calibration are presented. An overview of the database is presented in Sect. 4. A comparison against available satellite data and an assessment of uncertainties is given in Sect. 5. Concluding remarks are made in the last section.

## 2 The Antarctic Circumnavigation Expedition

ACE took place from 20 December 2016 to 19 March 2017, encompassing the entire Austral summer. It consisted of a voyage around the Southern Ocean between 34° and 74° S aboard the Russian research icebreaker *Akademik Tryoshnikov* (see technical details in Walton and Thomas, 2018).

The voyage was divided into three legs. Leg 1 was along the Indian Ocean from Cape Town, South Africa, to Hobart, Australia, with stops at Marion Island, Iles de Crozet et Kerguelen, and Heard Island. Leg 2 went across the Pacific Ocean to Punta Arenas, Chile, with stations at Mertz Glacier, Balleny Islands, Scott Island, Mount Siple (the southernmost station), Peter I Island, and Diego Ramírez. Leg 3 crossed the Atlantic Ocean back to Cape Town via South Georgia, South Sandwich Islands, and Bouvetøya. In addition, scientific observations were carried out during transit across the Atlantic on the way to/from South Africa (legs 0 and 4, respectively).

A schematic of the expedition is presented in Fig. 1 and photos of the environmental conditions are reported in Fig. 2. Leg 1 and Leg 3 mostly covered the open ocean north of the 60th parallel, roughly between the sub-Antarctic and polar fronts delimiting the Antarctic Circumpolar Current (Fig. 2a and b). Leg 2 primarily concentrated on the Antarctic marginal ice zone (see Fig. 2c) south of the 60th parallel, with two transects across the western Pacific Ocean sector south of Tasmania and the Drake Passage at the beginning and at the end of the leg. Average sea ice extent during ACE as detected by the Advanced Microwave Scanning Radiometer 2 sensor (AMSR2 – https://seaice.uni-bremen.de/sea-ice-concentration/amsre-amsr2/, last access: 15 June 2020; Spreen et al., 2008) is shown in Fig. 1.

## 3 The sea state monitoring system

### 3.1 Instrumentation and technical configuration

Sea state observations were recorded with the wave and surface current monitoring system WaMoS-II (details on software, hardware, and measurement principles can be found in Reichert et al., 1999). The instrument uses the marine X-

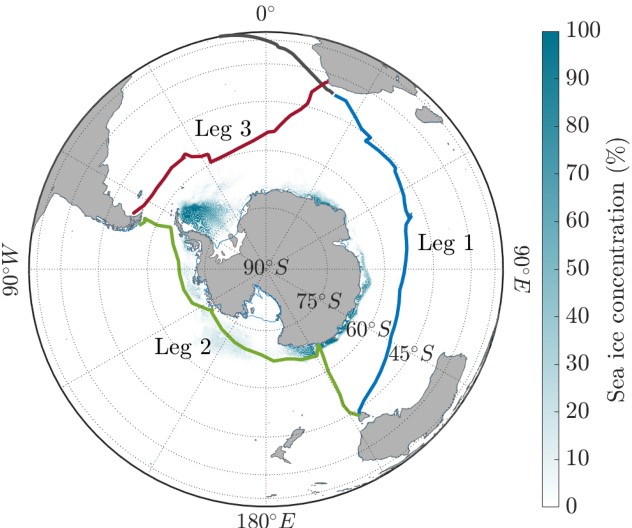

**Figure 1.** Map of the ACE voyage divided by legs. Average sea ice concentration during the expedition is also shown.

band radar (9.41 GHz) – a standard equipment on any vessels – to acquire high-definition radar images of the surrounding ocean surface and derive the directional wave energy spectrum, related integral parameters such as the significant wave height and mean wave period, and surface current speed and direction. Performance of WaMoS-II and its limitations are discussed in Hessner et al. (2002), Hessner et al. (2008), Hessner et al. (2019), Lund et al. (2015a), and Lund et al. (2015b). A summary of the range and accuracy of measured parameters is reported in Appendix A.

The overall system consists of an $A/D$ converter, a PC, and a processing software connected to the X-band radar (a schematic of WaMoS-II is presented in Fig. 3a). The basic configuration for the X-band radar requires an antenna with rotation speed of 24 rpm, horizontal opening angle of 0.9°, and radar pulse width of 80 ns. In addition, the radar has to be operated in the near range, i.e. 1.5 nautical miles ($\approx 2.8$ km). This allows WaMoS-II to acquire a radar image with a spatial resolution of 12 m and an angular resolution of 0.9° for every radar rotation (a sample image is reported in Fig. 3b). Further, water depth from the echo sounder, ship's positions, speed, and course from a Global Positioning System (GPS) receiver and true wind velocity and direction from two two-dimensional sonic anemometers operating as part of an automated weather station (AWS) and mounted at 31.5 m above mean sea level (see Schmale et al., 2019; Landwehr et al., 2020a; Thurnherr et al., 2020) are fed into the system. Wind measurements were acquired at a rate of 1 Hz, averaged over 175 s and converted from the measurement height to a neutral 10 m wind speed ($U_{10}$) by assuming a logarithmic profile (see Holthuijsen, 2007) before being passed on to the WaMoS-II.

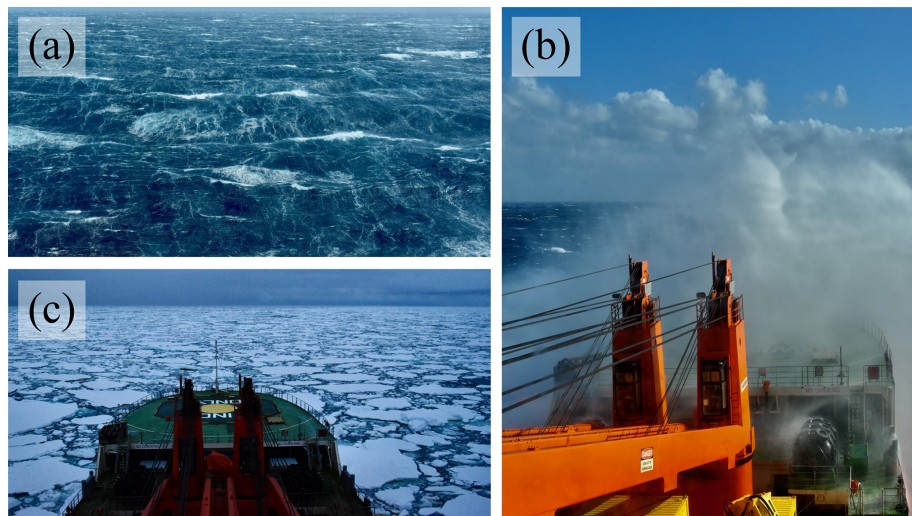

**Figure 2.** Examples of sea state conditions: ocean surface during storm conditions **(a)**, sailing through a storm **(b)**, and the marginal ice zone **(c)**.

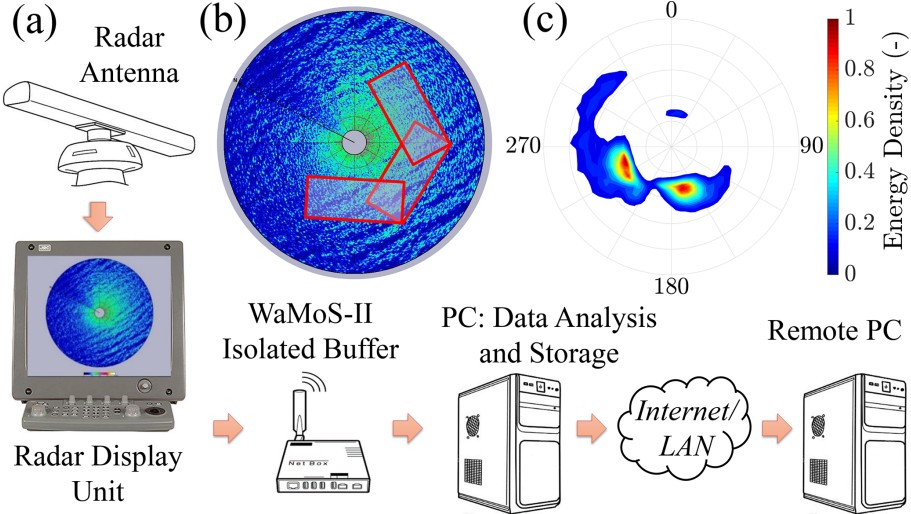

**Figure 3.** Schematic of the wave and surface current monitoring system WaMoS-II **(a)**, example of radar imagery and sub-areas (not in scale) for post-processing **(b)**, and a sample-derived directional wave energy spectrum **(c)**.

## 3.2    Measurement principles

The marine radar forms images of the surrounding area based on the backscatter of radar beams. The short wavelets (ripples) on the ocean surface contribute notably to reflection, while long-wave components modulate the returning signal. This results in stripe-like patterns in the radar images that correspond to the waves (these patterns are known as sea clutters). The system can detect sea clutters reliably only if wind speed is greater than $3\,\mathrm{m\,s^{-1}}$, which ensures the ocean surface is rough enough (i.e. ripples are developed) to backscatter the signal efficiently (Hatten et al., 1998). A small portion of the observations during ACE (approximately

9 %) were taken during low wind speed and hence removed from the data set.

The basic input for extracting sea state features is a sequence of 64 consecutive images, which correspond to a time period of 175 s (one complete image is acquired for every full turn of the antenna). Post-processing is carried out on sub-areas of $600\,\mathrm{m} \times 1200\,\mathrm{m}$ normally taken in front of the vessel, at port and at starboard, to avoid contamination due to the ship's wake (an example of sub-areas is presented in Fig. 3b). The temporal sequence for each sub-area, $I_s(x, y, t)$, is transformed with a three-dimensional discrete Fourier transform into its spectral domain counterpart; i.e. a three-dimensional image spectrum $I^{(3)}(k_x, k_y, \omega)$, where $\boldsymbol{k} = (k_x, k_y)$ is the two-dimensional wave number vector, and $\omega$ is the angu-

lar frequency. Assuming linear wave theory (Holthuijsen, 2007), spectral components in $I^{(3)}(k_x, k_y, \omega)$ that correspond to ocean waves have to satisfy the linear dispersion relation

$$\omega = \sqrt{g|\boldsymbol{k}|\tanh(|\boldsymbol{k}|d)} + \boldsymbol{k}\boldsymbol{u}, \tag{1}$$

where $g$ is the acceleration due to gravity, $d$ is the water depth, $\boldsymbol{u}$ is the surface current, and $|\boldsymbol{k}| = \sqrt{k_x^2 + k_y^2}$ is the wave number. Spectral components that do not obey Eq. (1) are assumed to be noise and hence removed. The remaining (filtered) three-dimensional image spectrum is integrated over the positive frequency domain to obtain a wave number image spectrum $I(k_x, k_y)$. The latter, however, does not coincide with the wave energy spectrum, because it represents the intensity of the radar backscatter rather than the amplitude of the water surface elevation (Nieto Borge et al., 1999; Hessner et al., 2002). Therefore, its zeroth-order moment ($m_0$) represents a signal-to-noise ratio (SNR) instead of the significant wave height, i.e. a measure of average wave height that is defined as $H_s = 4\sqrt{m_0}$. Consequently, the image spectrum requires a re-scaling to convert SNR into the corresponding wave height. This is achieved with the linear regression equation (see Nieto Borge et al., 1999, 2004)

$$H_s = A + B\sqrt{\mathrm{SNR}}, \tag{2}$$

where $A$ and $B$ are empirical constants that have to be calibrated following installation. Re-scaling $m_0$ enables correction of energy at each spectral mode and derivation of the wave energy spectrum $E_r(k_x, k_y)$. Radar imaging effects like tilt modulation, which refers to changes in the effective incidence angle along the long-wave slope, and shadowing, which is caused by the highest waves in the image, contribute to an inaccurate form of the resulting spectral density function, shifting energy towards high wave numbers (Nieto Borge et al., 2004). These effects depend on the view geometry (height and range of the antenna). Consequently, tilting and shadowing can be assumed to be homogeneous in the relatively small sub-areas used for post-processing and can be minimised with a single modulational transfer function (MTF, Nieto Borge et al., 2004). As the imaging effects depend on the wavelength, the MTF is a function of the wave number that corrects the spectral density at each mode. An ensemble average over all sub-areas is computed to derive the final wave spectrum $E(k_x, k_y)$ from the input 64 images.

In its standard output format, WaMoS-II archives the wave spectrum as a function of wave frequency, $f = \omega/2\pi$, and direction, $\vartheta - E(f, \vartheta)$; the change in variables from wave numbers to frequency–direction satisfies the dispersion relation in Eq. (1). An example of directional wave spectrum is shown in Fig. 3c. The resolution of the wave energy spectrum is dictated by the size of the sub-areas, which are used to derive the wave number counterpart in the first instance, and not by the temporal window. Considering the resolution of the image (12 m) and the minimum dimension of the sub-area (600 m),

WaMoS-II can detect wavelengths between 15 m and 600 m, which correspond to wave periods from 3 s to $\approx 16$ s.

Specific wave parameters are derived by integrating $E(f, \vartheta)$. These include the significant wave height, dominant and mean wave periods, associated wavelengths, directional width, and mean wave direction (see Appendix A for a full list of parameters and their definitions). WaMoS-II also partitions the directional wave energy spectrum to derive wave heights and periods for wind sea and the first three swell systems. The partitioning of the wave spectrum is performed using the path-of-steepest-ascent technique (Hanson and Phillips, 2001), which is a specific implementation of the inverse catchment scheme introduced by Hasselmann et al. (1996). The spectral peak that satisfies the condition

$$1.2 \frac{U}{c_p} \cos(\theta - \psi) > 1, \tag{3}$$

where $U$ is the wind speed, $c_p$ is the phase velocity, $\theta$ is the wave direction, and $\psi$ is the wind direction, is assumed to be associated with the wind sea. All other systems are swell and are ranked based on their energy contents as primary, secondary, and tertiary swell.

Ocean currents induce a Doppler shift to the wave field. Both current speed and direction can be quantified by minimising the distance between the position of the spectral energy in $I^{(3)}(k_x, k_y, \omega)$ and the theoretical position given by Eq. (1) using least-squares techniques (Young et al., 1985). The vessel's forward speed and heading are used to derive the true current.

Rain, snow, and sea ice produce an excess signal backscatter, which results in low-quality images and consequently inaccurate post-processing products. WaMoS-II automatically assesses the reliability of images through an internal quality control protocol (see Hessner et al., 2019), which evaluates backscatter intensity, number of sea clutters, and stability of ship motion among other parameters (we remark that tilting and shadowing effects are compensated for independently using the MTF and do not contribute to quality control). Images that are deemed of low quality are excluded. The majority of low-quality images were acquired in the marginal ice zone (i.e. south of the 60th parallel) during Leg 2. As a consequence, observations of waves in ice are not available in the present database.

## 3.3 Underway observations and file types

WaMoS-II operated continuously to record observations of the sea state during ACE. The vessel was equipped with one X-band radar, which was shared between science (requiring short range settings) and navigational aid (operating at medium and long range). Therefore, data acquisition was interrupted anytime the radar was needed for navigation, resulting in gaps in the observations. This was most common during Leg 2, as the radar was often switched to long range to detect icebergs.

https://doi.org/10.5194/essd-13-1-2021

Earth Syst. Sci. Data, 13, 1–21, 2021

The wave spectrum $E(f, \vartheta)$ was sampled at 175 s, assuming no gaps or corrupted images occur during the sampling of 64 consecutive radar images. Output files consist of (i) the directional wave energy spectrum in the wave number domain ($E(k_x, k_y)$, file extension D2S) and frequency–directional domain ($E(f, \vartheta)$, file extension FTH) and (ii) the (single) one-dimensional frequency–energy spectrum $S(f)$ obtained by integrating the directional spectrum $E(f, \vartheta)$ over $\vartheta$ (file extension D1S). Each file also includes a header that provides metadata such as geographical references (latitude and longitudes), time, wind speed and direction, ship speed and heading, current speed and direction, and additional integrated parameters.

WaMoS-II also performed a running average over 20 min to minimise the effects of natural variability. Output files consist of (i) the mean directional wave energy spectrum in both wave number and frequency–directional domain (file extensions D2M and FTM, respectively) and (ii) the mean one-dimensional frequency–energy spectrum (file extension D1M). These files are sampled every 175 s, with the first one 20 min after starting the equipment.

In addition, time series of wind, current, and wave variables from mean directional wave spectra are archived in monthly summary files.

## 3.4 Calibration

The calibration of coefficients $A$ and $B$ in Eq. (2) was performed by forcing the SNR to match independent benchmark observations of $H_s$. The reference values were reconstructed from records of ship motion, which were measured throughout the expedition with an inertial measurement unit (IMU) at a sampling rate of 1 Hz (Alberello et al., 2020b; Landwehr et al., 2021).

An overview of ship motion to sea state conversion (the wave buoy analogy) can be found in for example Nielsen (2017). The method relies on the principle that the vessel is a rigid body with six degrees of freedom (three translations: heave, surge, and sway; and three rotations: pitch, roll, and yaw) that moves in response to the incident wave field expressed as the frequency spectrum $S(f) = \int E(f, \vartheta) \, d\vartheta$ and restoring forces expressed as a function of its mass, geometry, loading conditions, and forward speed, among other parameters (Newman, 2018). The relation between the ship motion and the wave field is evaluated via the response amplitude operator ($R(f)$; see Newman, 2018), i.e. a ship-specific function that translates the motion spectrum $S_{\text{ship}}(f)$ into the wave spectrum: $S(f) = S_{\text{ship}}(f)/R(f)^2$. Motion spectra were evaluated by applying a discrete Fourier transform to 5 min long time series of heave motion. An approximation of $R(f)$ for the *Akademik Tryoshnikov* was calculated solving the equation of motion with a model based on the boundary element method (NEMOH, Babarit and Delhommeau, 2015) and taking into account the ship's heading, forward speed, and loading conditions. The model is based on a linear approach, and, thus, nonlinearities were excluded. The significant wave height was validated against freely available satellite altimeter data (Ribal and Young, 2019); the scatter plot of satellite observations versus reconstructed $H_s$ is presented in Fig. B1 of Appendix B.

Coefficients $A$ and $B$ in Eq. (2) were estimated using a maximum likelihood method for the period 9–11 December 2016 (Leg 0). The root mean square error (RMSE) of the fit is 0.21 m, with correlation coefficient $R \approx 0.90$ and scatter index SI $\approx 0.1$. Time series of $H_s$ derived from the IMU sensor and calibrated $H_s$ from WaMoS-II are shown in Fig. 4. Calibrated $A$ and $B$ coefficients were subsequently used to re-scale individual modes of the energy spectrum.

## 4 Overview of sea state conditions

### 4.1 Sea state climate during ACE

Excluding the regions south of the 60th parallel, which undergo a strong seasonal sea ice cycle (Eayrs et al., 2019), the Southern Ocean is normally characterised by weak seasonal variability (Young et al., 2020). Therefore, extreme sea states remain likely even during summer. As a reference, wind, current, and wave climate statistics in the form of the 50th and 90th percentiles (hereafter P50 and P90, respectively) aggregated in $2° \times 2°$ regions for the summer months (December, January, February) are reported in Fig. 5. Data of wind speed and wave height are from all satellite missions mounting altimeter sensors that are available from 1985 to 2019 (Ribal and Young, 2019). Data of current speed are from the Copernicus GlobCurrent database – https://marine.copernicus.eu (last access: 8 July 2020) – that combines the velocity field of geostrophic surface currents from satellite sensors recorded from 1993 to 2019 (Rio et al., 2014) and modelled Ekman currents, which include components from wind stress forcing obtained from atmospheric system and drifter data.

The wind speed is represented by its value at 10 m above sea level ($U_{10}$). Apart from a region east of Argentina, where the South American continent induces a shadowing effect, wind speed is fairly uniform throughout the ocean: P50 varies between 10 and 12 m s$^{-1}$, while P90 ranges between 15 and 18 m s$^{-1}$ (see Fig. 5a and b). Close to the Antarctic continent and outside the belt of the strong westerly winds (south of the 70th parallel), wind speed weakens with P50 reducing to $\approx 3$ m s$^{-1}$ and P90 to $\approx 10$ m s$^{-1}$. There are also low wind speeds ($U_{10} < 3$ m s$^{-1}$ for both P50 and P90) in the lee of the Antarctic Peninsula, although this may relate to uncertainties due to a high concentration of sea ice (see Fig. 1) and/or the increased drag over sea ice compared to open water (Martinson and Wamser, 1990). Note that, excluding the station at Mount Siple, the expedition remained within the belt of westerly winds.

Significant wave height $H_s$ follows the wind pattern, underpinning the dominance of wind seas on swell systems. Between 40° and 60° S, the belt where most of Leg 1 and

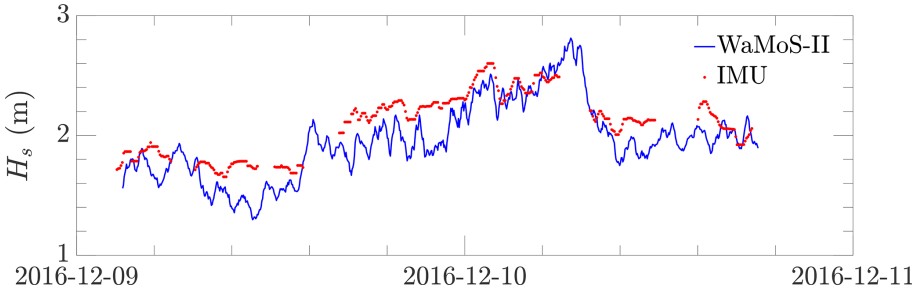

**Figure 4.** Time series of significant wave height $H_s$: benchmark observations derived from ship motion data (red dots) and calibrated records from WaMoS-II (blue solid line).

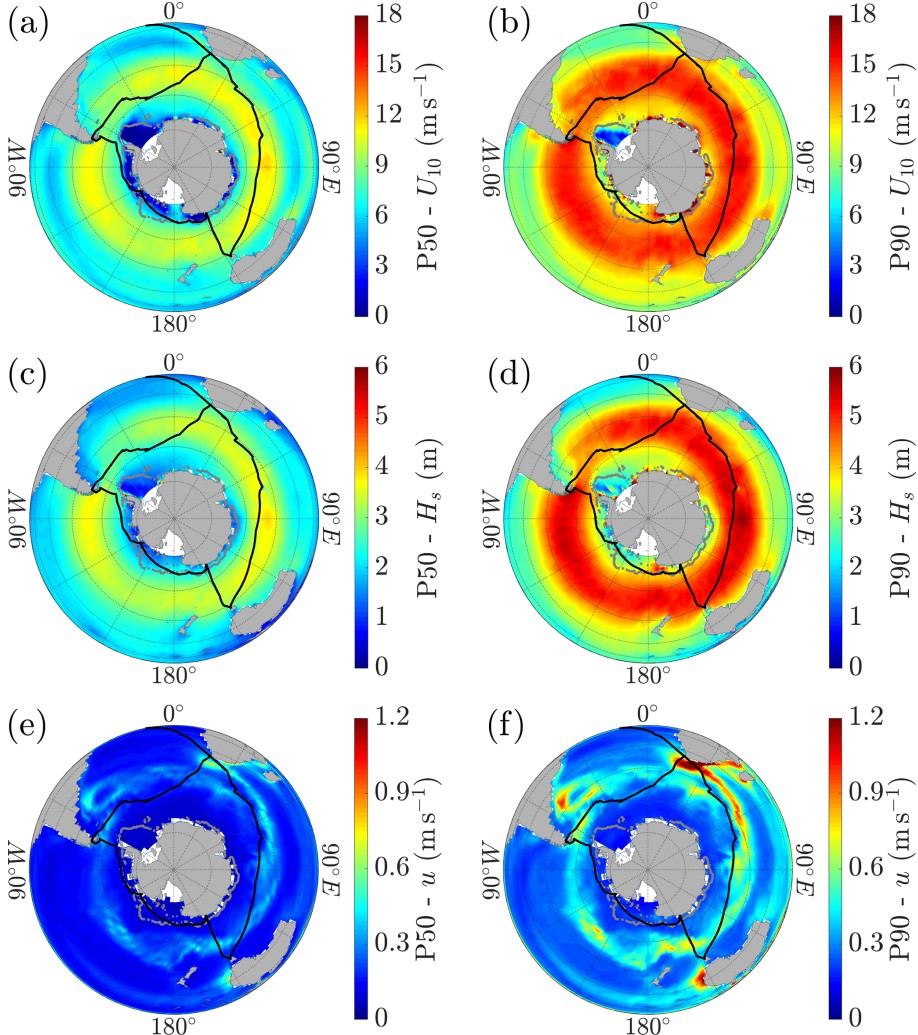

**Figure 5.** Wind speed ($U_{10}$), significant wave height ($H_s$), and surface current speed ($u$) climatology in Austral summer: **(a)** 50th percentile (median) wind speed, **(b)** 90th percentile wind speed, **(c)** 50th percentile (median) significant wave height, **(d)** 90th percentile significant wave height, **(e)** 50th percentile (median) surface current speed, and **(f)** 90th percentile surface current speed. Latitudes are shown every 15° (from 15 to 90° S) by thin lines; the route of the ACE voyage is reported as a black solid line; and the sea ice edge, defined by the 10 % sea ice concentration, is shown as a grey solid line.

Leg 3 took place, P50 $\approx$ 3.5 m, and P90 ranges between 5 and 6 m (Fig. 5c and d). There is an evident shadowing effect east of the Drake Passage, due to a combination of lower wind speed and a reduction of fetches. South of the 60th parallel (Leg 2), $H_s$ drops notably with the P50 decreasing to $\approx$ 2 m and P90 reducing to $\approx$ 4 m, despite strong westerly winds being active down to 70° S. The attenuation is induced by sea ice (Bennetts et al., 2015; Toffoli et al., 2015; Montiel et al., 2016), which has high concentration close to the Antarctic coastline, even in the summer months, especially in the western Pacific, Ross Sea and Amundsen Sea sectors of Antarctica (see Fig. 1).

The speed of the Antarctic Circumpolar Current has P50 $\approx$ 0.5 m s$^{-1}$. The P90 shows velocity with maxima in excess of 0.75 m s$^{-1}$, especially in the Indian Ocean sector (first half of Leg 1). Besides the Antarctic Circumpolar Current, ACE crossed two regions with strong currents: the Agulhas region, where P90 excess 1.5 m s$^{-1}$; and the Argentine basin (beginning of Leg 3), where a northward extension of the Antarctic Circumpolar Current forces water flow to speeds of approximately 1 m s$^{-1}$. South of the polar front (latitudes higher than 60° S), there is no significant circulation pattern, with maximum current speed less than 0.3 m s$^{-1}$ (see Fig. 5f).

## 4.2 Observed sea states during ACE

Figure 6 shows time series for the entire expedition of 10 m true wind speed ($U_{10}$), significant wave height ($H_s$), and current speed ($u$). As a benchmark, collocated values of P50, their interquartile range (IQR), and P90 are reported in the figure. Time series of mean wave period ($T_{m-10}$), wave directional spreading ($\sigma_\theta$), inverse wave age ($\mu$), and wave steepness ($\varepsilon$) are presented in Fig. 7. Definitions of the variables are reported in Appendix A.

Overall, the observed median wind speed was 7.25 m s$^{-1}$, with an interquartile (IQ) of 5.1 m s$^{-1}$. During Leg 1, the expedition went through six storm events with wind speeds reported in excess of 12 m s$^{-1}$ (P50 $\approx$ 10 m s$^{-1}$). Two of these events were equal to or greater than the P90 for the season ($\approx$ 15 m s$^{-1}$). Leg 2 started with the most extreme storm during ACE; winds reached speeds close to 20 m s$^{-1}$, which is well above P90. The remainder of Leg 2 was characterised by relatively low wind speeds, consistent with P50. Two more storms with wind speeds in excess of P90 were encountered at the end of Leg 2, while approaching and crossing the Drake Passage. The final leg was also characterised by intense storms with wind speeds notably above P50 for almost the entire leg. Three significant storm events with wind speeds above P90 ($U_{10} \approx$ 18 m s$^{-1}$) were reported.

The median significant wave height during the expedition was 2.61 m and IQ $\approx$ 1.6 m. To avoid the most energetic waves, the ship's course was continuously adapted to bypass storms. Despite this, intense wave conditions were encountered, with $H_s$ reaching the P90 ($\approx$ 5 m) during almost all storm events, especially during Leg 1 and Leg 3. The

largest waves ($H_s$ > 6 m) were encountered at the beginning of Leg 2 (see photos of the sea state in Fig. 2a and b). Thereafter, $H_s$ was less than 2 m as a result of the interaction with sea ice (see Fig. 2c). The crossing of the Drake Passage at the end of Leg 2 did not record significantly large waves, with $H_s \approx$ 4 m at most. Wave periods were generally long and normally in excess of 8 s (> 100 m wavelengths), substantiating the extensive (almost infinite) fetches for wave development. Concomitantly with almost all storms, $T_{m-10}$ increased and reached maximum values of 11–12 s (wavelengths $\approx$ 200 m).

The majority of Leg 1 and Leg 3 followed the Antarctic Circumpolar Current with records of surface current speeds oscillating around 0.5 m s$^{-1}$. Interestingly, observations were notably higher than P90 for the majority of the expedition. Despite being primarily south of the polar front, currents faster than P90 were also recorded in Leg 2, primarily in the marginal ice zone. ACE crossed two regions characterised by strong surface currents: east of South Africa at the southernmost edge of the Agulhas Current (beginning of Leg 1), with speeds up to 2 m s$^{-1}$; and east of South America where the Antarctic Circumpolar Current has a northward extension, with surface speeds recorded up to 1.8 m s$^{-1}$. Observations at both locations exceeded P90 notably. We remark that P50 and P90 include the contribution of geostrophic surface currents and wind stresses. However, additional components such as inertial oscillations (Treguier and Klein, 1994) are not taken into account due to the coarse resolution of satellite observations and Ekman components. To some extent, the absence of inertial oscillations in climate statistics substantiates the significant current speeds recorded by the WaMoS-II. Further, Ekman components remain uncertain in the Southern Ocean due to inaccuracies in estimating wind stress from the atmospheric system, adding inconsistencies to benchmark statistics.

An intrinsic feature of oceanic sea states is the directional distribution of the spectral density function (Mitsuyasu et al., 1975; Donelan et al., 1985; Young and Verhagen, 1996; Toffoli et al., 2017; Fadaeiazar et al., 2020; Young et al., 2020), which is summarised in the form of a mean directional spreading (i.e. the circular standard deviation of the directional wave energy spectrum). Sea states dominated by strong winds are normally characterised by a broad spreading with $\sigma_\vartheta$ > 40° (Donelan et al., 1985). These conditions were reported consistently throughout the expedition, with maximum mean directional spreading reaching values as large as 80°. Narrow directional distributions ($\sigma_\vartheta \leq$ 30°) were also common and primarily recorded in between storms, where the sea state was dominated by swells.

The inverse wave age $\mu$ and the wave steepness $\varepsilon$ are parameters that estimate the stage of growth of the wave field. Both variables are associated with nonlinear mechanisms that lead to large (extreme) waves (Onorato et al., 2009; Toffoli et al., 2017), wave breaking (Toffoli et al., 2010), and, thus, ocean–atmosphere fluxes (Schmale et al., 2019; Thurnherr et al., 2020). The inverse wave age is the ratio of wind

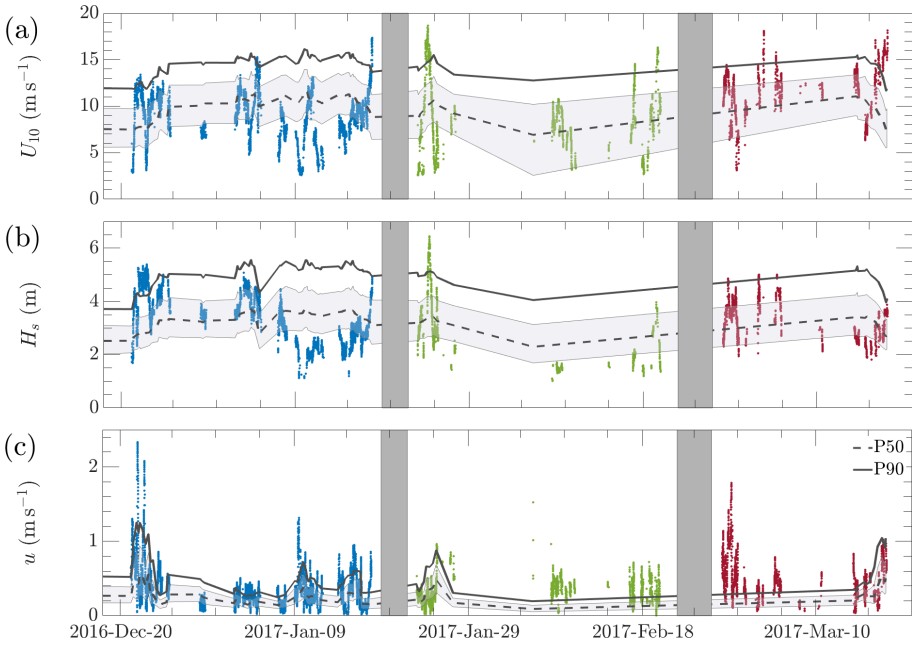

**Figure 6.** Time series of sea state variables in Leg 1 (blue), Leg 2 (green), and Leg 3 (red): **(a)** wind speed from the automated weather station, **(b)** significant wave height, and **(c)** current speed. For each variable, the dashed line and shading represent the 50th percentile and its interquartile range IQR, respectively, based on climate statistics from satellite observations; the solid line indicates the 90th percentile.

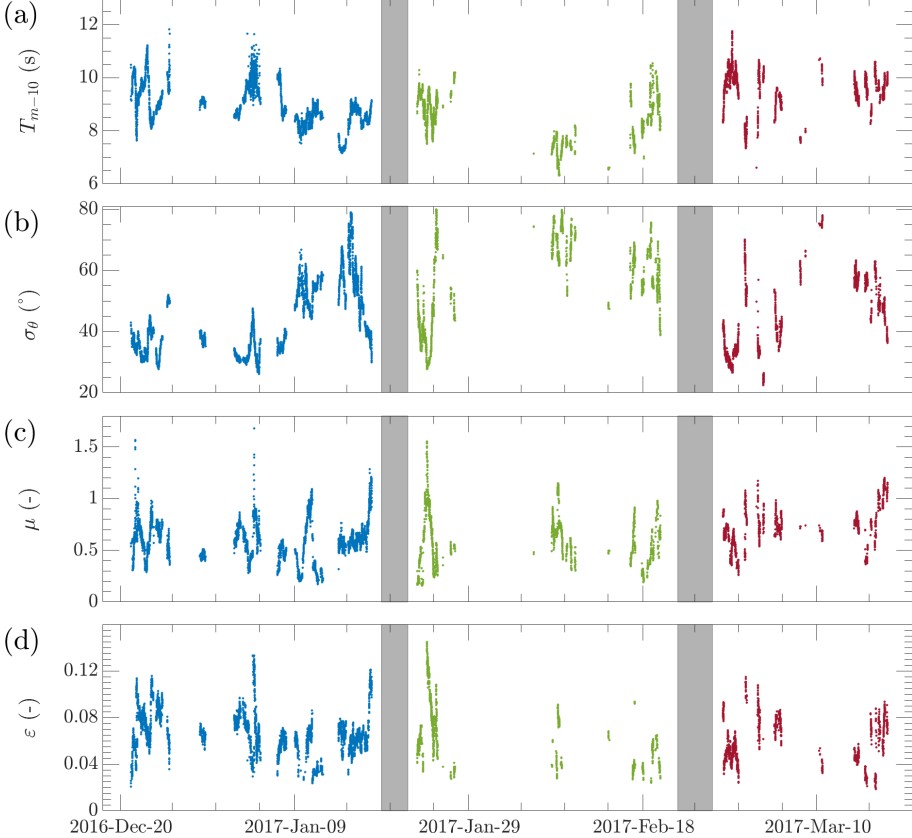

**Figure 7.** Time series of sea state variables in Leg 1 (blue), Leg 2 (green), and Leg 3 (red): **(a)** mean wave period, **(b)** mean directional spread, **(c)** inverse wave age, and **(d)** wave steepness. Details of variables are reported in Appendix A.

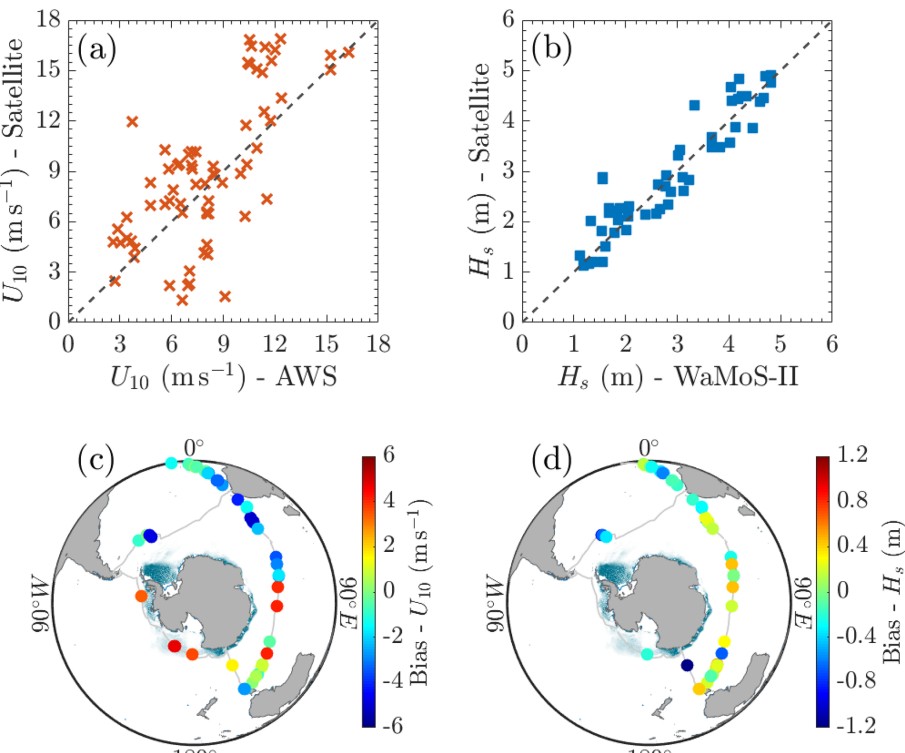

**Figure 8.** Wind from the automated weather station and significant wave height from WaMoS-II versus satellite observations: scatter diagrams **(a, b)** and geographical distribution of biases **(c, d)**. Average sea ice concentration during the expedition is overlaid in panels **(c)** and **(d)**.

speed to wave phase velocity (i.e. the ratio of wavelength to period). Following their generation, waves grow in height and length until they move faster than winds (Holthuijsen, 2007). For $\mu > 0.8$, waves are "young" as they are in a growing phase. This condition is normally characterised by a steep profile, which leads to breaking. Young waves were recorded during all storm events with steepness generally in excess of 0.1. For $\mu < 0.8$, the waves no longer receive energy from wind as they have reached full development. The shape of waves is gently sloping (i.e. the wave steepness is small) and breaking is unlikely (the ocean is dominated by swell). During the most extreme events at the beginning of Leg 2, steepness reached a maximum of about 0.13. This is an exceptionally high value for ocean waves and is normally associated with the formation of rogue waves (Onorato et al., 2009; Toffoli et al., 2010).

## 5 Comparison against satellite observations

### 5.1 Wind speed and significant wave height

Wind speeds and significant wave heights are compared against collocated satellite observations from altimeter sensors (same data source discussed in the previous session). Due to the scattered nature of satellite data, average values are computed for clusters with spatial resolution of $\Delta X = $

$\Delta Y = 0.5°$ and temporal resolution of $\Delta t = 3\,\text{h}$. Although satellite observations have been quality controlled and calibrated against available in situ sensors (see details in Ribal and Young, 2019), the scarcity of in situ data in the Southern Ocean leaves uncertainties in the data set.

Figure 8 shows scatter diagrams of matching averages at collocated clusters (panels a and b) and geographical distributions of biases (difference between WaMoS-II and satellite observations, panels c and d). Overall, in situ measurements of wind speeds during ACE are consistent with concurrent satellite observations, with data lying along the 1 : 1 correlation line. Nevertheless, there is a notable RMSE $\approx 3.2\,\text{m\,s}^{-1}$, with $R \approx 0.70$ and SI $\approx 0.360$. Biases show both overestimations (especially at the beginning of Leg 1 and Leg 3) and underestimation (at the end of Leg 1 and Leg 2) of satellite observations, varying between $-6$ and $6\,\text{m\,s}^{-1}$. The most substantial positive biases are reported in the marginal ice zone, where Antarctic sea ice affects wind speed detection.

Significant wave heights match better with satellite observations than wind speeds, with RMSE $\approx 0.42\,\text{m}$, $R \approx 0.93$, and SI $\approx 0.155$. Most of the collocated observations were found in Leg 1. Overall, the bias is positive, indicating a slight underestimation of the sea state from satellite sensors. The largest biases (ranging between 0.4 and 1.2 m) were

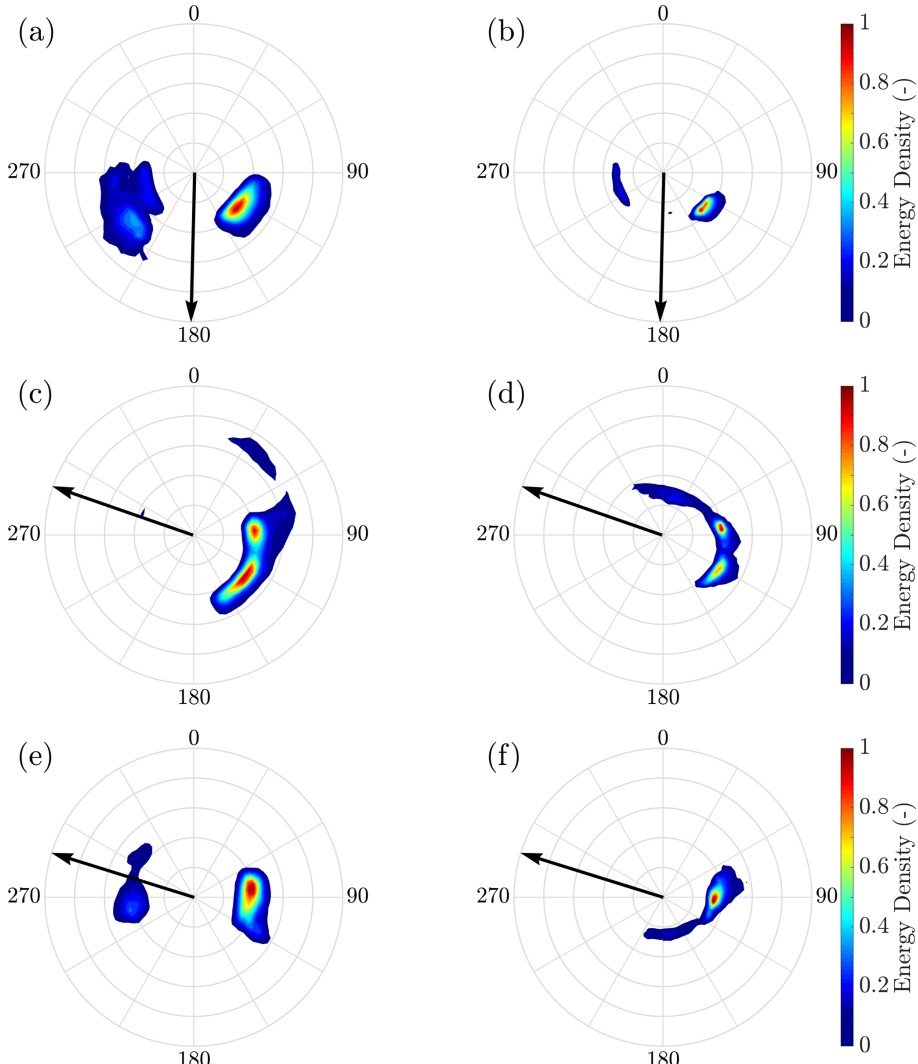

**Figure 9.** Example directional wave energy spectra recorded during ACE (**a, c, e**) and collocated SAR spectra (**b, d, f**). Wind direction recorded during ACE is shown as black arrows. Both the wave spectra and wind direction follow the "coming from" convention. Circles in the polar plot indicate frequencies from 0.05 Hz (innermost) to 0.25 Hz (outermost) with a step of 0.05 Hz; radiant lines indicate direction with a 30° step.

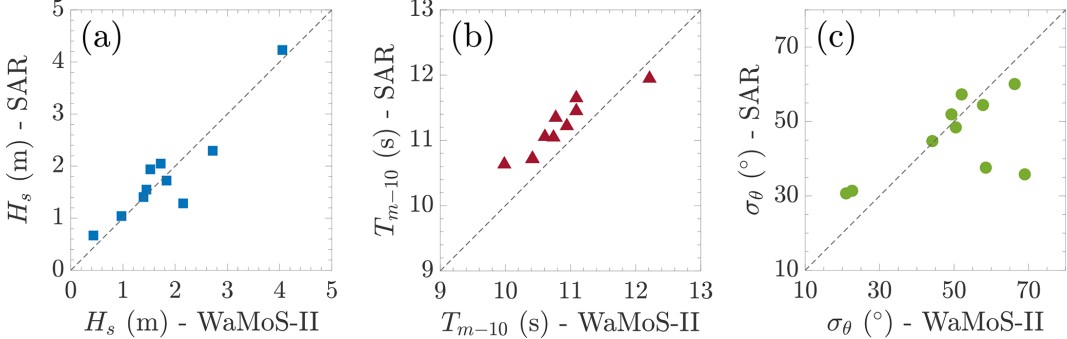

**Figure 10.** Comparison of integrated parameters (WaMoS-II versus SAR): significant wave height ($H_s$, panel **a**), energy wave period ($T_{m-10}$, panel **b**), and wave directional spreading ($\sigma_\theta$, panel **c**).

https://doi.org/10.5194/essd-13-1-2021 Earth Syst. Sci. Data, 13, 1–21, 2021

linked to storm events and are $\approx 10\%$ of the in situ-measured values.

## 5.2 Directional wave spectrum

Altimeter sensors only measure specific variables, namely the significant wave height and the wind speed, whereas SAR imagery can be converted into a directional wave energy spectrum (e.g. Collard et al., 2009). Collocated SAR spectra from Sentinel-1A/1B missions within area of $0.5° \times 0.5°$ and maximum temporal difference of 6 h were retrieved from the Australian Ocean Data Network (AODN) portal (Khan et al., 2020b). Overall, 10 SAR spectra were found during ACE, with $\approx 70\%$ of them in the Indian Ocean during Leg 1.

Examples of collocated wave spectra from WaMoS-II and SAR are presented in Fig. 9; mean wind direction is also reported. We remark that SAR detects wavelength longer than 115 m (approximately, wave periods exceeding 8 s or frequencies below 0.1 Hz) and represented swell systems primarily. WaMoS-II, on the contrary, captures the full spectrum, including the short wavelengths of the wind sea. Within the operational range of SAR ($f < 0.1$ Hz in the figure), the spectral shape from both sensors agrees well, especially for the portion around the primary (most energetic) swell. Notable discrepancies, however, are evident for less energetic secondary peaks, for which the relative uncertainty grows. High-frequency components ($f > 0.1$ Hz) are not resolved in SAR but appear in the WaMoS-II spectra. Note that the misalignment of high-frequency components with the wind direction in the upper two panels is due to recent wind change.

To provide a more robust comparison, scatter diagrams for $H_s$, $T_{m-10}$, and $\sigma_\vartheta$ are presented in Fig. 10. For consistency, wave spectra from WaMoS-II have been filtered to eliminate high-frequency modes that are not detected by SAR ($f > 0.117$ Hz or wavelength $L < 115$ m). SAR and WaMoS-II observations agree well, with RMSE $\approx 0.36$ m, $R \approx 0.92$, and SI $\approx 0.20$ for $H_s$; RMSE $\approx 0.42$ s, $R \approx 0.92$, and SI $\approx 0.038$ for $T_{m-10}$, noting wave periods from SAR are consistently (slightly) higher than WaMoS-II's; and RMSE $\approx 13.41°$, $R \approx 0.56$, and SI $\approx 0.295$ for $\sigma_\vartheta$, despite two outliers.

## 5.3 Surface current

Figure 11 shows the scatter diagram comparing surface current speeds from WaMoS-II and collocated observations derived from altimeter sensors (Rio et al., 2014). The geographical distributions of current speeds, directions, and concurrent differences between WaMoS-II and altimeter sensors are presented in Fig. 12. Note that values in both figures represent averages of observations falling in clusters of $0.5° \times 0.5°$ with temporal resolution of 0.5 h.

Contrary to wind and wave parameters, current speeds from WaMoS-II show notable differences from satellite observations. The former produces current speeds that are about

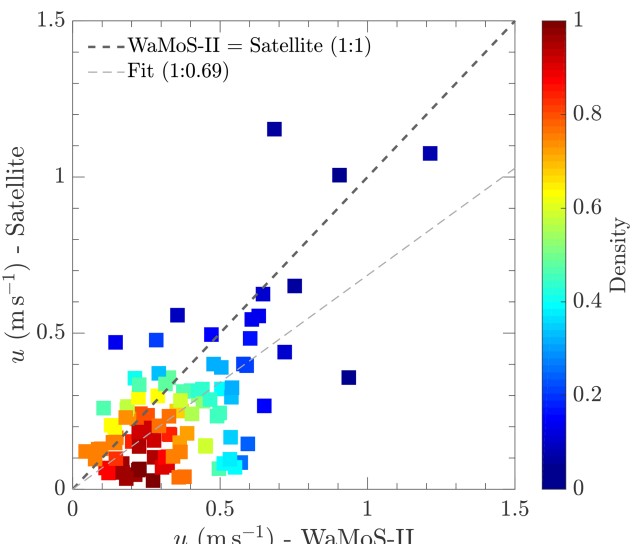

**Figure 11.** Scatter plot of WaMoS-II surface current speeds against observations derived from satellite sensors.

30 % larger than the latter. Other basic metrics of the scatter diagram are RMSE $\approx 0.2$ m s$^{-1}$, $R \approx 0.63$, and SI $\approx 0.80$. Biases associated with current speed are uniformly distributed across the expedition. A relatively small bias was detected in Leg 1, when sailing along the Antarctic Circumpolar Current. The largest biases (about 0.5 m s$^{-1}$, the same order of magnitude of the current speed itself) were detected primarily at the beginning of Leg 1 and at the end of Leg 3, where the ship crossed the Agulhas Current, and in Leg 2, when crossing the Antarctic marginal ice zone. The reported differences are linked to inconsistencies between WaMoS-II and benchmark data due to inertial oscillations (Treguier and Klein, 1994), which are not detected by satellite observations, and inaccuracy of wind stresses in the Ekman components.

Current direction is generally in better agreement with satellite observations than speed (see Fig. 12a and b). Differences between WaMoS-II altimeter sensors are normally small throughout the expedition, with common values of about 10°. The only substantial differences were recorded at the beginning of Leg 3, east of South America.

## 6 Data availability

Data are available through the Australian Antarctic Data Centre: (i) Alberello et al. (2020c) contains data sets of wave spectra including files D1S, D2S, D1M, D2M, and FTH (http://dx.doi.org/10.26179/5ed0a30aaf764); and (ii) Derkani et al. (2020) contains time series of wind speed and direction, current speed and direction, sea state parameters including wave height, period, wavelength and mean direction for total sea, wind sea and swell systems, ship course, position, and speed for each month of the expedition

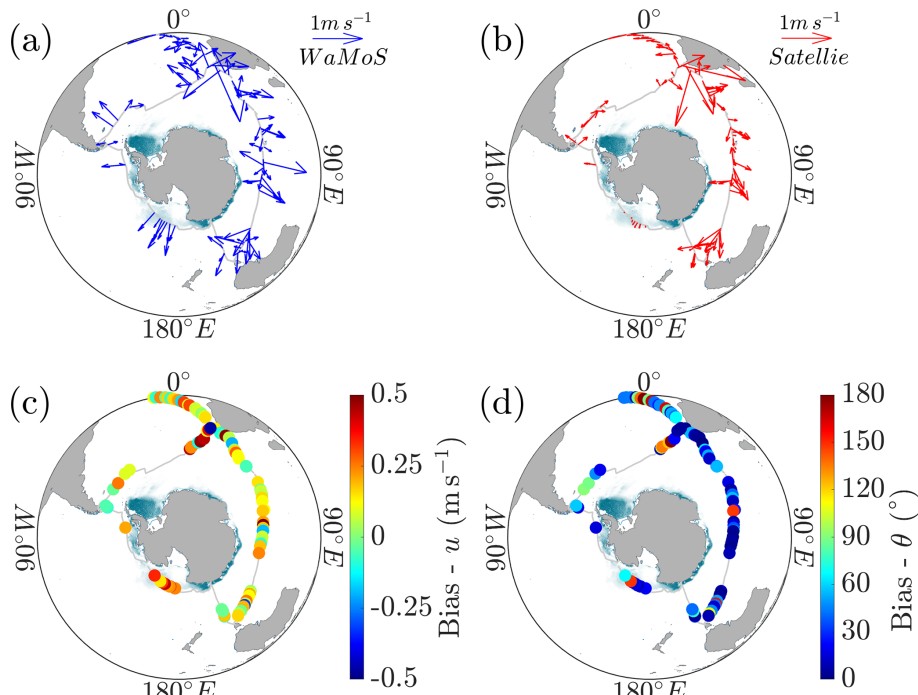

**Figure 12.** Surface current velocities along the ACE voyage: WaMoS-II **(a)** and space-borne altimeter sensors **(b)**. Surface current biases between WaMoS-II and satellite observations for current speed **(c)** and direction **(d)**. Average sea ice concentration during the expedition is overlaid in all panels.

(http://dx.doi.org/10.26179/5e9d038c396f2). Day and time of available measurements can be found in the file Available_Measurements_List.txt, which is included in Alberello et al. (2020c).

## 7 Conclusions

The scarcity of field observations in the Southern Ocean hampers the accuracy of satellite sensors and prediction models. In response to this issue, a unique data set of sea state parameters, comprising concomitant observations of winds, waves, and surface currents, was recorded in the Southern Ocean during the Antarctic Circumnavigation Expedition, from December 2016 to March 2017. Measurements were obtained using a radar-based wave and surface current monitoring system (WaMoS-II) and complemented with records of winds from the meteorological station on board the research icebreaker *Akademik Tryoshnikov*. Despite some gaps, observations of wind speeds and directions; directional wave energy spectra; integrated parameters such as wave heights, mean wave periods, and wavelengths; and current speeds and directions were collected underway during the entire expedition with outputs every 175 s. The sea state monitoring system was calibrated with benchmark sea state records, which were reconstructed from the ship motion. Measurements were also compared against available observations

from satellite-borne sensors to verify the robustness of the database.

The data set includes observations around the Southern Ocean from latitude 34 to 74° south. This comprises records in the open ocean across the Antarctic Circumpolar Current and in the Antarctic marginal ice zone. Due to its exposure to strong westerly winds, the Southern Ocean is subjected to harsh sea state conditions all year round. Although the expedition took place during the Austral summer, the data set contains records of severe sea states, in excess of the 90th percentile expected for the season.

The expedition was conceived to bring together a broad range of Earth Science disciplines with the aim of exploring the interplay of processes in the lower atmosphere, ocean surface, subsurface, and land with simultaneous observations. Additional data sets of atmospheric and oceanic variables that relate to sea state can be found in Schmale et al. (2019), Rodríguez-Ros et al. (2020), Smart et al. (2020), Thurnherr et al. (2020), and Suaria et al. (2020). These include, but are not limited to, air–sea fluxes (mass, gas, heat, and momentum), aerosol concentrations, stable water isotopologues, and micro-fibres. Collocated observations have been the foundation for research on moist diabatic processes (Thurnherr et al., 2020) and sea spray aerosols dynamics (Landwehr et al., 2021), demonstrating capacities for surface waves to modulate water isotopologue concentrations and marine aerosols emissions, settling velocity and lifetime in the ma-

rine boundary layer up to the cloud condensation height. The presented database has further potentials to support research enhancing wave model performances in the Southern Ocean, wave dynamics, including occurrence of rogue waves, wave dissipation mechanisms as well as other coupled processes, including those interconnecting waves with the upper ocean and sea ice in the Antarctic marginal ice zone.

## Appendix A: WaMoS-II sea state parameters

Details of sea-state-related variables from WaMoS-II output files as well as integrated parameters are described in Table A1. The $n$th-order moment of the spectral density function, $m_n$, referred to in the table is defined as $m_n = \int \int f^n E(f, \vartheta) \, \mathrm{d}f \, \mathrm{d}\vartheta$. Directional Fourier coefficients $a$ and $b$ used to compute the wave directional spreading are as follows:

$$a = \int \int \cos(\vartheta) S(f, \vartheta) \, \mathrm{d}f \, \mathrm{d}\vartheta, \tag{A1}$$

$$b = \int \int \sin(\vartheta) S(f, \vartheta) \, \mathrm{d}f \, \mathrm{d}\vartheta. \tag{A2}$$

https://doi.org/10.5194/essd-13-1-2021

Earth Syst. Sci. Data, 13, 1–21, 2021

**Table A1.** WaMoS-II output and integrated sea-state-related parameters and their symbol, definition, range, and accuracy. "n/a" stands for "not applicable".

| Sea-state-related parameter | Symbol | Definition | Range | Accuracy |
|---|---|---|---|---|
| 10 m true wind speed and direction (m s$^{-1}$, °) | $U_{10}, \alpha$ | – | – | – |
| Two-dimensional wave number spectrum (m$^4$) | $E(k_x, k_y)$ | Refer to Sect. 3.2 | – | n/a |
| Two-dimensional frequency–direction spectrum (m$^2$ (Hz × rad)$^{-1}$) | $E(f, \vartheta)$ | $\lvert k \rvert \frac{\partial \lvert k \rvert}{\partial \omega} E(k_x, k_y)$ | 0.0078–0.5000 Hz, 0–360° | n/a |
| One-dimensional frequency spectrum (m$^2$ Hz$^{-1}$) | $S(f)$ | $\int_0^{360°} E(f, \vartheta) \mathrm{d}\vartheta$ | 0.0078–0.5000 Hz | n/a |
| Significant wave height (m) | $H_S$ | Refer to Eq. (2) | 1–20 m | ±0.5 m |
| Energy wave period (s) | $T_{m-10}$ | $T_{m-10} = \frac{m_{-1}}{m_0}$ | 3.5–55 s | ±0.5 s |
| Peak wave period (s) | $T_p$ | $\frac{1}{f_p}$ | 3.5–55 s | ±0.5 s |
| Mean wave direction (°) | $\beta_m$ | $\arctan(b/a)$ | 0–360° | ±2° |
| Peak wave direction (°) | $\beta_p$ | $\vartheta(f_p); f_p = \frac{1}{T_p}$ | 0–360° | ±2° |
| Peak wave length (m) | $\lambda_p$ | $\lambda_p = \frac{g T_p^2}{2\pi} \sqrt{\tanh(\frac{4\pi^2}{T_p^2} \frac{d}{g})}$ | 19–600 m | – |
| First, second, and third significant wave height for swell systems 1, 2, and 3 (m) | $H_{s1,2,or3}$ | $4\sqrt{m_{0_{1,2,or3}}}$ | 1–20 m | ±0.5 m |
| First, second, and third wave peak period for swell systems 1, 2, and 3 (s) | $T_{p1,2,or3}$ | $\frac{1}{f_{p1,2,or3}}$ | 3.5–55 s | ±0.5 s |
| First, second, and third wave length for swell systems 1, 2, and 3 (m) | $\lambda_{1,2,or3}$ | $\frac{2\pi}{\lvert k_{p1,2,or3} \rvert}$ | 19–600 m | – |
| First, second, and third wave direction for swell systems 1, 2, and 3 (°) | $\beta_{1,2,or3}$ | $\vartheta(f_{p1,2,or3})$ | 0–360° | ±2° |
| Wave directional spreading (°) | $\sigma_\theta$ | $\sqrt{2[1 - \sqrt{\frac{a^2+b^2}{m_0^2}}]}$ TS1 | 0–90° TS2 | n/a |
| Inverse wave age (−) | $\mu$ | $\frac{U_{10}}{c_p}$, $c_p$: wave phase velocity | – | – |
| Wave steepness (−) | $\varepsilon$ | $k\frac{H_S}{2}$ | – | – |
| Surface current speed (m s$^{-1}$) | $u$ | Refer to Eq. (1) | 0–20 m s$^{-1}$ | ±0.2 m s$^{-1}$ |
| Surface current direction (°) | $\theta$ | Refer to Eq. (1) | 0–360° | ±2° |

https://doi.org/10.5194/essd-13-1-2021

## Appendix B: Validation of ship motion to sea state conversion

Significant wave heights reconstructed from ship motion data were validated against freely available satellite observations (Ribal and Young, 2019) for the entire ACE voyage (see scatter plot in Fig. B1). Due to the coarse resolution of satellite data, average values are computed for clusters with spatial resolution of $0.5° \times 0.5°$ and temporal resolution of 3 h. Overall, there is a good agreement between reconstructed and observed sea state. The root-mean squared error (RMSE) is 0.4 m, the correlation coefficient ($R$) is 0.94, and the scatter index (SI) is 0.17. Similar error metrics are obtained by comparing the reconstructed sea state against parameters from the European Centre for Medium-Range Weather Forecasts (ECMWF – https://www.ecmwf.int/en/forecasts/datasets/reanalysis-datasets/era5, last access: 7 December 2020) ERA-5 database.

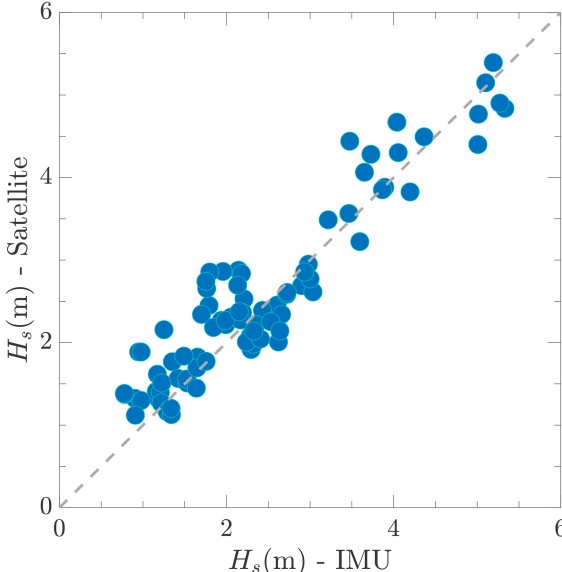

**Figure B1.** Satellite observations versus significant wave height reconstructed from ship motion.

**Author contributions.** KR, KMH, and AT participated to the expedition and acquired the data. MHD, AA, LGB, and AT conceived the manuscript. KGH provided on-shore technical support and LA provided marine forecast during the expedition. MHD and KGH calibrated and analysed the data. FN provided benchmark observation for calibration. LA and SK provided SAR spectra. All authors contributed to the data interpretation and to the writing of the manuscript.

**Competing interests.** The authors declare that they have no conflict of interest.

**Acknowledgements.** This work was part of the Antarctic Circumnavigation Expedition (ACE). MHD was partially supported by a PhD top-up scholarship from the Australian Bureau of Meteorology. MHD, AA, FN and AT acknowledge technical support form the Air-Sea-Ice Lab initiative.

**Financial support.** This research has been supported by the ACE Foundation and Ferring Pharmaceuticals (Project 17), the CRC-P initiative of the Australian Government (grant no. CRC- P53991), and the Australian Antarctic Program (grant no. AAS 4434).

**Review statement.** This paper was edited by Giuseppe M.R. Manzella and reviewed by three anonymous referees.

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

## Remarks from the typesetter

TS1    Please give an explanation of why this needs to be changed. We have to ask the handling editor for approval. Thanks.

TS2    Please give an explanation of why this needs to be changed. We have to ask the handling editor for approval. Thanks.