# Peer review of "Wind, waves, and surface currents in the Southern Ocean: Observations from the Antarctic Circumnavigation Expedition"

_Earth System Science Data, 2020_

## Referee Comment (RC1) · Anonymous Referee #1 · 22 Oct 2020

This manuscript presents an observation data set of surface wind, surface waves and surface currents obtained during a 3-month oceanographic expedition in the Southern Ocean. The most original part of this data set was obtained by analyzing marine radar observations (radar WAMOS_II) carried out from the research vessel.

It is a very good initiative to publish the details on this data set. Indeed, first, the number of local observations in this part of the oceans is very scarce, and there are not so many oceanographic research cruises. This hampers many scientific studies focused on this region and more generally studies in conditions of high wind and high sea-states. More generally, field observations of surface wind, waves and surface current remain very im-

portant to progress on several topics related to the air/sea interface as : - better understanding or quantifying physical processes related to surface ocean waves (wave/wave interactions, wave/current interactions, wave/ice interactions, impact of wave on turbulence and ocean/atmosphere fluxes, - improving numerical modelling (wave models and/or coupled atmospheric/wave/oceanographic models), - validating and improving satellite products on wind, waves and current, particularly in extreme wind and wave conditions.

So it is very likely that the data will be used in future by scientist not involved themselves in this field campaign.

The manuscript is well organized and provides the main information to future users of the data set. Maybe, as suggested by the topic editor, more information could be given in the abstract and conclusion on the questions already addressed by the PIs of these measurements, and those that could be addressed in the future by external users.

In general, the methods and materials are well described. Some details are lacking but can be easily added (see below specific comments). References to instrumental design and processing methods are also pertinent (except some, see, below specific comments).

I checked, on some examples, that the data files are accessible and well documented. There are two documentation pages associated to the DOIs and an easy access to the data files through a structure in directoryÂă/sub_directory/files organized by dates. Maybe a general calendar could be added so that a user can see immediately if data sets exist on their dates of interest. Also, one information which I could not find is: do you include somewhere in your data sets, the information on the sea-ice cover? (could be interesting if available)

A validation of the data set is presented in the manuscript, at least for what concerns the wave height (comparison with satellite data). For the other parameters, due to the lack of concomitant independent observations, I do understand that the validation remains limited. However, I suggest to add here some references to previous publications on WAMOS –II data sets to let the reader know what are the expected performances or known limitations on other parameters of the data set such as dominant wave direction, dominant frequency, directional spread, surface current.

Overall, my recommendation, taking into account the specificity of the ESSD Journal and its focus on original research data sets furthering the reuse of high-quality data, is to accept this manuscript , provided that some minor revisions are carried out, to answer my specific comments below.

See attached document for specific and detailed comments

Please also note the supplement to this comment:
https://essd.copernicus.org/preprints/essd-2020-255/essd-2020-255-RC1-supplement.pdf

**Supplement:**

Comments on the manuscript entitled « Wind, waves, and surface currents in the Southern Ocean:
Observations from the Antarctic Circumnavigation Expedition »,
submitted by Derkani et al,
to Earth System Science Data, oct 2020.

**General comments**

This manuscript presents an observation data set of surface wind, surface waves and surface currents obtained during a 3-month oceanographic expedition in the Southern Ocean. The most original part of this data set was obtained by analyzing marine radar observations (radar WAMOS_II) carried out from the research vessel.

It is a very good initiative to publish the details on this data set. Indeed, first, the number of local observations in this part of the oceans is very scarce, and there are not so many oceanographic research cruises. This hampers many scientific studies focused on this region and more generally studies in conditions of high wind and high sea-states. More generally, field observations of surface wind, waves and surface current remain very important to progress on several topics related to the air/sea interface as :
- better understanding or quantifying physical processes related to surface ocean waves (wave/wave interactions, wave/current interactions, wave/ice interactions, impact of wave on turbulence and ocean/atmosphere fluxes,
- improving numerical modelling (wave models and/or coupled atmospheric/wave/oceanographic models),
- validating and improving satellite products on wind, waves and current, particularly in extreme wind and wave conditions.

So it is very likely that the data will be used in future by scientist not involved themselves in this field campaign.

The manuscript is well organized and provides the main information to future users of the data set. Maybe, as suggested by the topic editor, more information could be given in the abstract and conclusion on the questions already addressed by the PIs of these measurements, and those that could be addressed in the future by external users.

In general, the methods and materials are well described. Some details are lacking but can be easily added (see below specific comments). References to instrumental design and processing methods are also pertinent (except some, see, below specific comments).

I checked, on some examples, that the data files are accessible and well documented. There are two documentation pages associated to the DOIs and an easy access to the data files through a structure in directory /sub_directory/files organized by dates. Maybe a general calendar could be added so that a user can see immediately if data sets exist on their dates of interest. Also, one information which I could not find is: do you include somewhere in your data sets, the information on the sea-ice cover? (could be interesting if available)

A validation of the data set is presented in the manuscript, at least for what concerns the wave height (comparison with satellite data). For the other parameters, due to the lack of concomitant independent observations, I do understand that the validation remains limited. However, I suggest to add here some references to previous publications on WAMOS –II data sets to let the reader know what are the expected performances or known limitations on other parameters of the data set such as dominant wave direction, dominant frequency, directional spread, surface current.

Overall, my recommendation, taking into account the specificity of the ESSD Journal and its focus on original research data sets furthering the reuse of high-quality data, is to accept this manuscript , provided that some minor revisions are carried out, to answer my specific comments below.

**Specific comments**

**-** section 3.1, line 97: more details should be added on the type of wind sensor, its position on the vessel, the height measurement, the calibration procedure

- line 120: please give more details on how the shadowing effects and tilting effects are removed. Is it a correction of a filter based on data quality control? How many data sets are eliminated by this procedure?

-line 124 and following: the method for rescaling the wave spectrum deserves more details. Indeed, I could not find details on this rescaling in the Young et al, 1985 publication. Furthermore, other publications on WAMOS, like the one of Nieto Borge et al, 2004 mention that this type of rescaling may not be fully appropriate, as the Transfer Function between image intensity and wave heights depends on the wave number of the ocean waves. Could you comment on that in the manuscript?

- line 132-133: please give details or references on how the partitions were estimated (method of partitioning, external data used in the partitioning if any - like wind speed and wind direction,…)

- section 3.4: I am surprised that only ship data are used to build reference values of significant wave height Hs. You do not have any possible comparison with buoy data when the ship was in coastal regions? Using ship IMU data as reference to obtain Hs does not seem so trivial as shown for example by Nielsen and Dietz (see e.g. "Estimation of sea-state parameters by the wave buoy analogy with comparisons spectral wave models », Ocean Engineering 2020) . In the ship to wave spectral transformation, do you take into account the possible non-linearities of the ship response, the effects of ship speed, of direction of waves with respect to the ship heading,….? More details should be added in this section. On the other hand, I must admit that the a posterior validation using satellite significant wave heights, as presented in Fig.7, is convincing

- Section 4.1 comments about the statistics on current: You have omitted to mention that the current from satellite altimeters are not surface currents but geostrophic currents.

- section 4.2? lines 204-207: please , indicate how the raw wind measurements were converted into ten meter-height winds (U10), and what is the duration of integration of the raw data.

- line 223-224: it is strange that the only references that you give to mention the oceanic directional distribution of waves come from wave tank measurements. Could you add some references on field measurements?

- line 240: here again , mention that the current measurements from WAMOS-II and the climatological currents estimated from altimeter data do not represent exactly the same geophysical quantity.

- lines 270-273: you could mention that on these examples, SAR does not detect the wind sea, in opposite to WAMOS-II data.

**Technical corrections**

- Figure 5: you could mention in the legend that the circles in dashed light lines (hardly visible) are plotted every 15° in latitude

- Line 201: "pattern" (instead of "patter")

- Figure 8 i) the marks for the scales are not visible (circles in wave number or frequency) ii)  Also could you add the wind direction on these polar plots

---

## Referee Comment (RC2) · Anonymous Referee #2 · 28 Oct 2020

This paper describes a dataset of winds, surface currents and ocean waves collected during the ACE expedition that circumnavigated the Southern Ocean from December 2016 to March 2017. The ocean wave observations from the WaMos-II wave and surface current monitoring system are particularly novel and the fact that the sea state and wind observations were made at the same time make these an interesting and extremely useful dataset for scientists studying air-sea-ice-ocean processes in the Southern Ocean. Such datasets are rare in the Southern Ocean and tend to be collected along regular shipping tracks. The datasets comprise both oceanic and atmospheric conditions over the three to four months and these novel data constitute a valuable resource that will be of interest to a variety of readers.

[Figure]

The manuscript is well-written and well-structured. The authors provide a good overview and introduction that synthesizes the literature on air-sea-ice-ocean processes in the Southern Ocean. They provide the rationale for collecting and publishing these data by highlighting the lack of in situ observations for the region and the consequences of the lack of observational data. The figures are well-presented, informative, and easy to follow. The manuscript supports the publication of these data sets.

The datasets are available and accessible at the links referenced in the abstract and under the "Data availability" heading. At these links, the datasets are well-described and straightforward to access.

I have some very minor comments detailed in the supplement pdf.

Please also note the supplement to this comment:
https://essd.copernicus.org/preprints/essd-2020-255/essd-2020-255-RC2-supplement.pdf

**Supplement:**

**General comments**

This paper describes a dataset of winds, surface currents and ocean waves collected during the ACE expedition that circumnavigated the Southern Ocean from December 2016 to March 2017. The ocean wave observations from the WaMos-II wave and surface current monitoring system are particularly novel and the fact that the sea state and wind observations were made at the same time make these an interesting and extremely useful dataset for scientists studying air-sea-ice-ocean processes in the Southern Ocean. Such datasets are rare in the Southern Ocean and tend to be collected along regular shipping tracks. The datasets comprise both oceanic and atmospheric conditions over the three to four months and these data constitute a valuable resource that will be of interest to a variety of readers.

The manuscript is well-written and well-structured. The authors provide a good overview and introduction that synthesizes the literature on air-sea-ice-ocean processes in the Southern Ocean. They provide the rationale for collecting and publishing these data by highlighting the lack of in situ observations for the region and the consequences of the lack of observational data. The figures are well-presented, informative, and easy to follow. The manuscript supports the publication of these data sets.

The datasets are available and accessible at the links referenced in the abstract and under the "Data availability" heading. At these links, the datasets are well-described and straightforward to access.

**I have some very minor comments detailed below:**

**Line 106-107:** Do you have an estimate of what proportion of the data (if any?) is not processed due to the ocean being too smooth? Is this much of an issue in the Southern Ocean?

**LIne 128:** "modes" = "mode"?

**Line 180:** is there a standard deviation or other measure you could include to show the variability of these variables over the 20-30 year time period? (This might be useful to

include in Fig 6 to put your instantaneous observations in context with the interannual variability?)

**Figure 5**: perhaps add a contour showing the sea ice edge to the figures that show wave height to highlight the attenuation you mention in line 195? (Figure 5c and/or Figure 5d).

**Line 202**: "patter" = "patterns"

**Line 202:** maybe add either the polar front or the legs to the figures showing surface currents to highlight the region you are describing here? It would be useful to reference Figure 5f here.

**Figure 7:** c and/or d - add the sea ice edge (or include the shading from Fig 1) so we can visualize where the sea ice is complicating estimations.

**Line 283:** Is there any other literature on WaMoS-II surface current observations that provides any assessment of the quality of the current observations? I.e. are you able to say anything about the quality of the WaMoS-II vs altimeter current measurements? How different are the current estimates - maybe it isn't appropriate to directly compare these quantities?

---

## Referee Comment (RC3) · Anonymous Referee #3 · 30 Nov 2020

The paper "Wind, waves, and surface currents in the Southern Ocean: Observations from the Antarctic Circumnavigation Expedition" by Derkani et al. presents the data collected during a 3-leg campaign circumnavigating Antarctica, between 2016 an 2017. The paper is well written, well structured and provides all the campaign information to complement the data, together with a coincise but comprehensive description of the Southern Ocean physical oceanography and its observing systems. More specific information on the data structure and file organisation can be found in the description of the data at the repositories referred to in the manuscript (accessible).

I recommend the paper is accepted. I only have few comments I would like the authors

consider:

- I would like a brief comment by the authors on the spectral frequency resolution and parameters variability that originates from the 160-second long duration of Wamos-II acquisition (if I correctly understood). I mean, with 160-s long records the spectral resolution over frequency is very low (large delta f). And given the rotation speed of the antenna I guess also the maximum resolved frequency may be very low. How does this affect the spectral representation? In addition, with Tm > 8 s (also 13 s) waves every sample includes less than 20 waves. So, I suspect the estimate of the spectrum and wave parameters (even including the 600x1200 m2 area) might be pretty unstable.

- wind data are measured by an onboard meteorological station, but in Figure 7 the measured wind U10 is labeled as Wamos-II. Please may you check consistency?

- In Figure 8, axis labels (units and variables) are missing.

---

## Author Comment (AC1) · 9 Jan 2021

**Response to Anonymous Reviewer #1**

*R#1:* As suggested by the topic editor, more information could be given in the abstract and conclusion on the questions already addressed by the PIs of these measurements, and those that could be addressed in the future by external users.

AC: Research supported by these data sets as well as future applications have been discussed in the abstract and conclusions.

**R#1:* A general calendar could be added so that a user can see immediately if data sets exist on their dates of interest.**

AC: We thank the reviewer for this suggestion. We have added a general calendar as an additional file (*Available\_Measurements\_List.txt*) to the data set. This new file provides day and time of available measurements in the format yyyy-mm-dd hh:MM:ss. It is also available via the link below:

https://u.pcloud.link/publink/show?code=XZjMtQXZjzHChPvK1Tyn9WGYLaGhsFWPpN67.

**R#1:* Also, one information which I could not find is: do you include somewhere in your data sets, the information on the sea-ice cover? (could be interesting if available)**

AC: Our observations are limited to open water conditions as measurements in the marginal ice zone were not accurate and thus excluded. We added a comment at the end of Section 3.2 to clarify the exclusion of data in sea ice. Information on sea ice cover is presented in Figure 1 of the manuscript and is retrieved from the Advanced Microwave Scanning Radiometer 2 (AMSR2) database. This database is publicly available at https://seaice.uni-bremen.de/sea-ice-concentration/amsre-amsr2/. We have included the link in the revised version.

*R#1: I suggest to add here some references to previous publications on WAMOS –II data sets to let the reader know what are the expected performances or known limitations on other parameters of the data set such as dominant wave direction,dominant frequency, directional spread, surface current.*

AC: We thank the reviewer for this suggestion. We have added a brief discussion on the performance of WAMOS-II and relevant references (Hassner et al., 2002; 2007; 2019; Lund et al., 2015b; 2015b) in Section 3.1.
**Specific comments**

**R#1:* Section 3.1, line 97: more details should be added on the type of wind sensor, its position on the vessel, the height measurement, the calibration procedure**

AC: Details on the wind sensors are extensively discussed in Landwehr et al. (2020), Schmale et al. (2019) and Thurnherr et al. (2020). A brief summary describing the wind sensor has been added in Section 3.1 for completeness.

**R#1:* Line 120: please give more details on how the shadowing effects and tilting effects are removed. Is it a correction of a filter based on data quality control? How many data sets are eliminated by this procedure?**

AC: To minimize imaging effects like shadowing and tilt modulation, the WaMoS-II wave analysis was carried out in areas within a limited range. This approach assumes that these effects are homogeneous and can be compensated by a single Modulation Transfer Function (MTF) as described in Nieto Borge et al. (2004). Note that this is a standard method in WaMoS-II and further details are not disclosed by the manufacturer and thus excluded. A brief discussion on mitigation of tilting and shadowing effects with the MTF (Nieto Borge et al., 2004) is added in Section 3.2.

The internal real time WaMoS-II quality control (iQC) is independent of the applied MTF. The iQC is based on various different individual tests (see Hessner et al., 2019), which evaluate different conditions required for reliable radar-based wave measurements (e.g. sufficient sea clutter information or stable ship motion conditions). Most of the data that were labeled as unreliable were recorded in ice-covered waters with no significant surface waves present or the radar not operating in required short pulse mode. Therefore, data in the marginal ice zone has not been included in the data set. This has been remarked in the revised version of the manuscript.
*R#1:* Line 124 and following: the method for rescaling the wave spectrum deserves more details. Indeed, I could not find details on this rescaling in the Young et al, 1985 publication. Furthermore, other publications on WAMOS, like the one of Nieto Borge et al, 2004 mention that this type of rescaling may not be fully appropriate, as the Transfer Function between image intensity and wave heights depends on the wave number of the ocean waves. Could you comment on that in the manuscript?

AC: We thank the reviewer for this comment. The rescaling that we used is the standard method implemented in the WaMoS-II software. Available details can be found in Nieto Borge et al. (1999) and Nieto Borge et al. (2004). These details are already summarised in the original manuscript (lines 124 and following). Note that additional, more detailed, information about the rescaling is not disclosed by the manufacturer and thus cannot be presented.

The reviewer is right when stating that no information is provided by Young at al. 1985 in this regard. Reference to that paper was included by mistake and the correct citations have been added to the revised version of the manuscript.

The capabilities of the rescaling technique are discussed and demonstrated in Nieto Borge et al. (1999) and Hessner et al. (2002); these two references have been added to the revised version of the manuscript and briefly commented on. We do not find any discussion on the appropriateness of rescaling techniques in Nieto Borge et al. (2004) as mentioned by the reviewer. If we misunderstood the comment, we would appreciate receiving more details in order to better address this concern. Nevertheless, our understanding of the discussion in Nieto Borge et al. (2004) is that the Modulation Transfer Function (MTF) compensates for the nonlinearities related to imaging effects such as tilt modulation and shadowing. As these depend on the view geometry (antenna height and range), WaMoS-II limits the analysis range to an area where the imaging effects are assumed to be homogeneous, allowing application of the MTF method. As the
imaging effects also depend on the wavelength, the MTF is a function of wavenumber. While the MTF allows extrapolating the wave spectrum from the image spectrum, the resulting spectral density still requires a calibration to return the correct energy content (the MTF returns the relative energy distribution and not the absolute one). In order to get the absolute values, the spectrum needs to be rescaled, so that the individual frequency and direction bins are related to wave energy. To do so, WaMoS-II uses the rescaling techniques discussed at lines 124 and following. The text has been reworded to clarify the overall process.

**R#1:* Line 132-133: please give details or references on how the partitions were estimated (method of partitioning, external data used in the partitioning if any - like wind speed and wind direction,...)**

AC: The partition of the wave spectrum is performed using the "path of steepest ascent" technique proposed by Hanson Phillips (2001), which is a specific implementation of the inverse catchment scheme introduced by Hasselmann et al. (1996). Different wave systems (i.e. spectral peaks or subcatchments) are determined by associating each spectral grid point to the neighbor with the highest energy level. Grid points corresponding to the same local peak are clustered, and each of these clusters defines a partition (watershed algorithm). The spectral peak that satisfies the condition

 $1.2\frac{U}{c_p}\cos(\theta - \psi) > 1,$

where U is the wind speed,  $c_p$  is the phase velocity,  $\theta$  is the wave direction and  $\psi$  is the wind direction, is associated with the wind sea. All other systems are swell and are ranked based on their energy contents as primary, secondary and tertiary swell. Details on partitioning and related references have been added to the revised version of the manuscript in Section 3.2.

**ESSDD**
*R#1:* Section 3.4: I am surprised that only ship data are used to build reference values of significant wave height Hs. You do not have any possible comparison with buoy data when the ship was in coastal regions? Using ship IMU data as reference to obtain Hs does not seem so trivial as shown for example by Nielsenand Dietz (see e.g. "Estimation of sea-state parameters by the wave buoy analogy with comparisons spectral wave modelsÂż, Ocean Engineering 2020). In the ship to wave spectral transformation, do you take into account the possible non-linearities of the ship response, the effects of ship speed, of direction of waves with respect to the ship heading,....? More details should be added in this section. On the other hand, I must admit that the a posterior validation using satellite significant wave heights, as presented in Fig.7, is convincing.

AC: That is correct, we could not perform a calibration based on buoy data because there were no co-located buoy measurements during ACE. Therefore, the calibration had to rely on the sea state retrieved from ship motion data. The underlying principle for sea state reconstruction is that the ship is a rigid body with six degrees of freedom (three translations: heave, surge, and sway; and three rotations: pitch, roll, and yaw) and moves in response to the incident wave field and restoring forces as a function of its mass, geometry, loading conditions and forward speed among other parameters (Newman, 2018). The relation linking ship motion and energy spectrum of the incident wave field is evaluated by the response amplitude operator (R(f), see Newman, 2018), i.e. a ship-specific function that translates the motion spectrum ( $S_{ship}(f)$ ) into the incident wave spectrum ( $S_{wave}(f)$ ):  $R(f)^{-2} = S_{wave}(f)/S_{ship}(f)$ .

Motion spectra were evaluated by applying a Fourier Transformation over 5 minute long time series of heave motion. R(f) was calculated by solving the equation of motion with a model based on boundary element methods (Babarit Delhommeau, 2015), taking into account the ship's heading, forward speed and loading conditions; the model is based on a linear approach and thus nonlinearities were excluded. The significant wave height was validated against freely available satellite altimeter data (Ribal Young,
2019) of significant wave height for the entire voyage. The text describing the sea state reconstruction from ship motion has been updated (see Section 3.4). Furthermore, we have added an appendix (Appendix B) in the revised manuscript, where we discuss the accuracy of the reconstructed sea state against satellite altimeter observations. The appendix includes Figure B1, which shows the scatter plot of significant wave height from satellite altimeter versus significant wave height reconstructed from the ship motion. Due to the coarse resolution of satellite data, average values are computed for clusters with spatial resolution of  $0.5^{\circ} \times 0.5^{\circ}$  and temporal resolution of 3 hours. There is a good agreement overall. The root-mean squared error (RMSE) is  $\approx 0.4$  m, the correlation coefficient (R) is  $\approx 0.94$ , and the scatter index (SI) is  $\approx 0.17$ . Note that similar error metrics have been obtained by comparing the reconstructed sea state against parameters from ECMWF ERA-5 reanalysis.

**R#1:* Section 4.1 comments about the statistics on current: You have omitted to mention that the current from satellite altimeters are not surface currents but geostrophic currents.**

AC: We thank the reviewer for spotting this error. We used current data from COPERNICUS-GLOBCURRENT - https://marine.copernicus.eu, which combines the total velocity field based on satellite geostrophic surface currents with modelled Ekman currents, which includes wind stress forcing obtained from atmospheric system and drifters data. Information on COPERNICUS-GLOBCURRENT has been added in Section 4.1.

**R#1:* Section 4.2 lines 204-207: please , indicate how the raw wind measurements were converted into ten meter-height winds (U10), and what is the duration of integration of the raw data.**

AC: Wind measurements (20-minutes average) are converted from the measurement height to a 10-metre above sea level wind speed  $(U_{10})$  by assuming a logarithmic profile
(see Landwehr at al., 2020). Furthermore, atmospheric boundary layer instability was not considered and thus  $U_{10}$  represents the neutral wind speed. Details have been added in Section 3.1.

**R#1:* Line 223-224: it is strange that the only references that you give to mention the oceanic directional distribution of waves come from wave tank measurements. Could you add some references on field measurements?**

AC: References to field measurements by Mitsuyasu et al. (1975), Donelan et al. (1985), Young et al. (1996) and Young et al. (2020) have been added.

**R#1:* Line 240: here again , mention that the current measurements from WAMOS-II and the climatological currents estimated from altimeter data do not represent exactly the same geophysical quantity.**

AC: The reviewer is right. Even though we used a combination of geostrophic surface currents and modelled Ekman currents, WaMoS-II still detects additional components such as inertial oscillations (Treguier and Klein, 1994), which are not represented in the benchmark data set. As inertial oscillations are particularly strong in the Southern Ocean, they represent a notable source of uncertainty. Furthermore, Ekman components remain uncertain in the Southern Ocean due to inaccuracies in estimating wind stress from the atmospheric system, adding more inconsistencies between benchmark and our observations. In the revised version of the manuscript we have commented on the differences between our observations and the benchmark data in Section 4.2.

**R#1: Lines 270-273: you could mention that on these examples, SAR does not detect the wind sea, in opposite to WAMOS-II data.**

AC: We thank the reviewer for this comment. We have added an additional statement to stress that SAR does not detect wind sea.
**Technical corrections**

**R#1:* Figure 5: you could mention in the legend that the circles in dashed light lines (hardly visible) are plotted every $15^{\circ}$ in latitude**

AC: We have edited the figure and added this information in the caption.

**R#1: Line 201: "pattern"(instead of "patter")**

AC: This typo has been corrected.

**R#1:* Figure 8 i) the marks for the scales are not visible (circles in wave number or frequency) ii) Also could you add the wind direction on these polar plots?**

AC: The figure has been updated accordingly.
References:

Babarit, A. and Delhommeau, G., 2015, September. Theoretical and numerical aspects of the open source BEM solver NEMOH.

Donelan, M.A., Hamilton, J. and Hui, W., 1985. Directional spectra of wind-generated ocean waves. Philosophical Transactions of the Royal Society of London. Series A, Mathematical and Physical Sciences, 315(1534), pp.509-562.

Hanson, J.L. and Phillips, O.M., 2001. Automated analysis of ocean surface directional wave spectra. Journal of atmospheric and oceanic technology, 18(2), pp.277-293.

Hasselmann, S., Brüning, C., Hasselmann, K. and Heimbach, P., 1996. An improved algorithm for the retrieval of ocean wave spectra from synthetic aperture radar image spectra. Journal of Geophysical Research: Oceans, 101(C7), pp.16615-16629.

Hessner, K.G., El Naggar, S., von Appen, W.J. and Strass, V.H., 2019. On the reliability of surface current measurements by X-Band marine radar. Remote Sensing, 11(9), p.1030.

Hessner, K., Reichert, K., Dittmer, J., Borge, J.C.N. and Günther, H., 2002. Evaluation of WaMoS II wave data. In Ocean Wave Measurement and Analysis (2001) (pp. 221-230).

Hessner, K., Nieto-Borge J.C., and Bell P.S., 2007. Nautical Radar Measurements in Europe, Applications of WaMoS II as a Sensor for Sea State, Current and Bathymetry. In: Sensing of the European Seas, Barale, Vittorio; Gade, Martin (Eds.), Springer, p435-446.

Landwehr, S., Thurnherr, I., Cassar, N., Gysel-Beer, M. and Schmale, J., 2020. Using global reanalysis data to quantify and correct airflow distortion bias in shipborne wind
speed measurements. Atmospheric Measurement Techniques, 13(6), pp.3487-3506.

Lund, B., Graber, H.C., Hessner, K. and Williams, N.J., 2015a. On shipboard marine X-band radar near-surface current "calibration". Journal of Atmospheric and Oceanic Technology, 32(10), pp.1928-1944.

Lund, B., Graber, H.C., Tamura, H., Collins III, C.O. and Varlamov, S.M., 2015b. A new technique for the retrieval of near-surface vertical current shear from marine X-band radar images. Journal of Geophysical Research: Oceans, 120(12), pp.8466-8486.

Mitsuyasu, H., Tasai, F., Suhara, T., Mizuno, S., Ohkusu, M., Honda, T. and Rikiishi, K., 1975. Observations of the directional spectrum of ocean WavesUsing a cloverleaf buoy. Journal of Physical Oceanography, 5(4), pp.750-760.

Newman, J.N., 2018. Marine hydrodynamics (p. 448). The MIT press.

Nielsen, U.D. and Dietz, J., 2020. Estimation of sea state parameters by the wave buoy analogy with comparisons to third generation spectral wave models. Ocean Engineering, 216, p.107781.

Schmale, J., Baccarini, A., Thurnherr, I., Henning, S., Efraim, A., Regayre, L., Bolas, C., Hartmann, M., Welti, A., Lehtipalo, K. and Aemisegger, F., 2019. Overview of the Antarctic circumnavigation expedition: Study of preindustrial-like aerosols and their climate effects (ACE-SPACE). Bulletin of the American Meteorological Society, 100(11), pp.2260-2283.

Nieto Borge, J., Hessner, K. and Reichert, K., 1999, July. Estimation of the significant wave height with X-band nautical radars. In Proc. 18th Int. Conf. Offshore Mechanics and Arctic Engineering (OMAE).
Nieto Borge, J., RodrÍguez, G.R., Hessner, K. and González, P.I., 2004. Inversion of marine radar images for surface wave analysis. Journal of Atmospheric and Oceanic Technology, 21(8), pp.1291-1300.

Ribal, A. and Young, I.R., 2019. 33 years of globally calibrated wave height and wind speed data based on altimeter observations. Scientific data, 6(1), pp.1-15.

Thurnherr, I., Kozachek, A., Graf, P., Weng, Y., Bolshiyanov, D., Landwehr, S., Pfahl, S., Schmale, J., Sodemann, H., Steen-Larsen, H.C. and Toffoli, A., 2020. Meridional and vertical variations of the water vapour isotopic composition in the marine boundary layer over the Atlantic and Southern Ocean. Atmospheric Chemistry and Physics, 20(ARTICLE), pp.5811-5835.

Treguier, A.M. and Klein, P., 1994. Instability of wind-forced inertial oscillations. Journal of Fluid Mechanics, 275, pp.323-349.

Young, I.R., Fontaine, E., Liu, Q. and Babanin, A.V., 2020. The Wave Climate of the Southern Ocean. Journal of Physical Oceanography, 50(5), pp.1417-1433.

Young, I.R., Rosenthal, W. and Ziemer, F., 1985. A three-dimensional analysis of marine radar images for the determination of ocean wave directionality and surface currents. Journal of Geophysical Research: Oceans, 90(C1), pp.1049-1059.

Young, I.R. and Verhagen, L.A., 1996. The growth of fetch limited waves in water of finite depth. Part 2. Spectral evolution. Coastal Engineering, 29(1-2), pp.79-99.

---

## Author Comment (AC2) · 9 Jan 2021

**Response to Anonymous Reviewer #2**

*R#2: Line 106-107: Do you have an estimate of what proportion of the data (if any?) is not processed due to the ocean being too smooth? Is this much of an issue in the Southern Ocean?*

AC: Although it is unusual for the Southern Ocean, there were conditions of very low wind speed, resulting in a smooth surface. In more general sense, WaMoS-II cannot detect the ocean surface accurately if wind speed is lower than $3\,m/s$. Low wind speed

affected about 9% of the whole observation. We commented on the low wind speed issue in the revised version of the manuscript in Section 3.2.

*R#2: Line 128: "modes" = "mode"?*

AC: This typo has been corrected.

*R#2: Line 180: Is there a standard deviation or other measure you could include to show the variability of these variables over the 20-30 year time period? (This might be useful to include in Fig 6 to put your instantaneous observations in context with the inter annual variability?)*

AC: We thank the reviewer for this comment. We have added error bars to climate statistics. To make the error bars more visible, we split the original figure 6 into two figures in the revised manuscript. Figure 6 shows variables for which climate statistics are available; Figure 7 shows all other variables.

*R#2: Figure 5: perhaps add a contour showing the sea ice edge to the figures that show wave height to highlight the attenuation you mention in line 195? (Figure 5c and/or Figure 5d).*

AC: We updated Figure 5 by adding the contour line of the sea ice edge as located at 10% ice concentration.

*R#2: Line 202: "patter" = "patterns"*

AC: This typo has been corrected.

*R#2: Line 202: maybe add either the polar front or the legs to the figures showing surface currents to highlight the region you are describing here? It would be useful to reference Figure 5f here.*

AC: Voyage's route has been added in Figure 5.

*R#2: Figure 7: c and/or d - add the sea ice edge (or include the shading from Fig 1) so we can visualize where the sea ice is complicating estimations.*

AC: Sea ice has been added to panels c and d of this figure. Note that figure 7 in the original manuscript is figure 8 in the revised version.

*R#2: Line 283: Is there any other literature on WaMoS-II surface current observations that provides any assessment of the quality of the current observations? I.e. are you able to say anything about the quality of the WaMoS-II vs altimeter current measurements? How different are the current estimates - maybe it isn't appropriate to directly compare these quantities?*

AC: We thank the reviewer for this comment. The accuracy of the ocean current measurements is discussed in Hessner et al. (2019), Lund et al. (2015a), and Lund et al. (2015b). A brief comment about accuracy has been added to the revised version of the manuscript (Section 3.1).

Regarding satellite observations, we used current data from COPERNICUS-GLOBCURRENT - https://marine.copernicus.eu, which provides the total velocity field based on satellite geostrophic surface currents and modelled Ekman currents, which take into account wind stress forcing obtained from atmospheric system and drifters data. In principle, this product is consistent with current measurements from WaMoS-II. Nevertheless, WaMoS-II also detects inertial oscillations (Treguier and Klein, 1994), which can be particularly intense in the Southern Ocean. These components are not

captured in the benchmark data and hence represent a source of inconsistency. Furthermore, Ekman components remain uncertain in the Southern Ocean due to inaccuracies in estimating wind stress from the atmospheric system, adding more inconsistencies between benchmark and our observations. A remark in this regard has been added to the manuscript in Section 5.3.
**References**

Hessner, K.G., El Naggar, S., von Appen, W.J. and Strass, V.H., 2019. On the reliability of surface current measurements by X-Band marine radar. Remote Sensing, 11(9), p.1030.

Lund, B., Graber, H.C., Hessner, K. and Williams, N.J., 2015a. On shipboard marine X-band radar near-surface current "calibration". Journal of Atmospheric and Oceanic Technology, 32(10), pp.1928-1944.

Lund, B., Graber, H.C., Tamura, H., Collins III, C.O. and Varlamov, S.M., 2015. A new technique for the retrieval of near‐surface vertical current shear from marine X‐band radar images. Journal of Geophysical Research: Oceans, 120(12), pp.8466-8486.

Treguier, A.M. and Klein, P., 1994. Instability of wind-forced inertial oscillations. Journal of Fluid Mechanics, 275, pp.323-349.
* * *

---

## Author Comment (AC3) · 9 Jan 2021

**Response to Anonymous Reviewer #3**

*R#3: I would like a brief comment by the authors on the spectral frequency resolution and parameters variability that originates from the 160-second long duration of Wamos-II acquisition (if I correctly understood). I mean, with 160-s long records the spectral resolution over frequency is very low (large delta f). And given the rotation speed of the antenna I guess also the maximum resolved frequency may be very low. How does this affect the spectral representation? In*

*addition, with Tm > 8 s (also 13 s) waves every sample includes less than 20 waves. So, I suspect the estimate of the spectrum and wave parameters (even including the 600×1200 m2 area) might be pretty unstable.*

AC: The X-band radar provides spatio-temporal information by recording 64 images of the surrounding ocean surface over a period of 175 s (we erroneously indicated 160 s in the original submission). This corresponds to one image for each full turn of the antenna. A single wave spectrum is computed from all images recorded within this period of time. Specifically, the spectral density is estimated by applying a two dimensional Fast Fourier Transform to three sub-areas (x,y) of dimensions 600 m × 1200 m for each image/surface. The final spectrum is computed as an ensemble average over the 175 s and all sub-areas. Therefore, the spectral resolution is dictated by the spatial resolution of the sub-area and not the temporal duration of the sampling. Considering the resolution of the image (5 m) and the minimum dimension of the sub-area (600 m), WaMoS-II can detect wavelengths between 10 m and 600 m which correspond to frequencies from 3 s to about 16 s. We have added a comment on the spectral resolution in the revised version of the manuscript (Section 3.2).

*R#3: Wind data are measured by an on board meteorological station, but in Figure 7 the measured wind U10 is labeled as Wamos-II. Please may you check consistency?*

AC: We thank the reviewer for spotting this error. Label of wind speed has been corrected in the revised version.

*R#3: In Figure 8, axis labels (units and variables) are missing.*

AC: The figure has been updated.

---

## Author Response (AR1)

**Response to Anonymous Reviewer #1**

***R#1: As suggested by the topic editor, more information could be given in the abstract and conclusion on the questions already addressed by the PIs of these measurements, and those that could be addressed in the future by external users.***

**AC:** Research supported by these data sets as well as future applications have been mentioned in the abstract and conclusions.

**Changes in manuscript:** We have modified the last sentences of the abstract at lines 11 to 15. The new text reads as

*"The data set is the most extensive and comprehensive collection of observations of surface processes for the Southern Ocean and is intended to underpin improvements of wave prediction models around Antarctica and research of air–sea interaction processes, including gas exchange and dynamics of sea spray aerosol particles. The data set has further potentials to support theoretical and numerical research on lower atmosphere, air–sea interface and upper ocean processes."*

The last paragraph of the conclusion has been edited (lines 364 to 374). The new text is as follows:

*"The expedition was conceived to bring together a broad range of Earth Science disciplines with the aim of exploring the interplay of processes in the lower atmosphere, ocean surface, subsurface and land with simultaneous observations. Additional data sets of atmospheric and oceanic variables that relate to sea state can be found in Schmale et al. (2019); Rodríguez-Ros et al. (2020); Smart et al. (2020); Thurnherr et al. (2020) and Suaria et al. (2020). These include, but are not limited to, air–sea fluxes (mass, gas, heat and momentum), aerosol concentrations, stable water isotopologues and micro-fibres. Collocated observations have been the foundation for research on moist diabatic processes (Thurnherr et al., 2020) and sea spray aerosols dynamics (Landwehr et al., 2020b), demonstrating capacities for surface waves to modulate water isotopologues concentrations and marine aerosols emissions, settling velocity and lifetime in the marine boundary layer up to the cloud condensation height. The presented database has further potentials to support research enhancing wave model performances in the Southern Ocean, wave dynamics, including occurrence of rogue waves, wave dissipation mechanisms as well as other coupled processes, including those interconnecting waves with the upper ocean and sea ice in the Antarctic marginal ice zone."*

***R#1: A general calendar could be added so that a user can see immediately if data sets exist on their dates of interest.***

**AC:** We thank the reviewer for this suggestion. We have added a general calendar as an additional file (*Available_Measurements_List.txt*) to the data set. This new file provides day and time of available measurements in the format yyyy-mm-dd hh:MM:ss. It is also available via the link below:
`https://u.pcloud.link/publink/show?code=XZjMtQXZjzHChPvK1Tyn9WGYLaGhsFWPpN67`.

**Changes in manuscript:** The following lines have been added at the end of Section 7:

*"Day and time of available measurements can be found in the file Available_Measurements_List.txt, which is included in* Alberello et al. (2020).*"*

***R#1: Also, one information which I could not find is: do you include somewhere in your data sets, the information on the sea-ice cover? (could be interesting if available)***

AC: Our observations are limited to open water conditions as measurements in the marginal ice zone were not accurate and thus excluded. We added a comment at the end of Section 3.2 to clarify the exclusion of data in sea ice. To highlight areas covered by sea ice, the geogrphical distribution of its concentration is presented in Figure 1 of the manuscript. Sea ice concentrations are from the Advanced Microwave Scanning Radiometer 2 (AMSR2) database, which is publicly available at `https://seaice.uni-bremen.de/sea-ice-concentration/amsre-amsr2/`. We have included the link in the revised version.

**Changes in manuscript:** We have modified the last paragraph of section 3.2 (lines 165 to 171) to mention that observation in sea ice are excluded. The new text reads as

*"Rain, snow and sea ice produce an excess signal backscatter, which results in low quality images and consequently inaccurate post processing products. WaMoS-II automatically assesses the reliability of images through an internal quality control protocol (see Hessner et al., 2019), which evaluate backscatter intensity, number of sea clutters and stability of ship motion among other parameters (we remark that tilting and shadowing effects are compensated independently using the MTF and do not contribute to quality control). Images that are deemed of low quality are excluded. The majority of low quality images were acquired in the marginal ice zone (i.e. south of the $60^{th}$ parallel) during Leg 2. As a consequence, observations of waves-in-ice are not available in the present database."*

An average sea ice coverage during the expedition, in the form of concentration, is shown in Figures 1, 8 and 12. Source for sea ice data is provided at line 90.

***R#1: I suggest to add here some references to previous publications on WAMOS −II data sets to let the reader know what are the expected performances or known limitations on other parameters of the data set such as dominant wave direction, dominant frequency, directional spread, surface current.***

**AC:** We thank the reviewer for this suggestion. We have added relevant references (Hessner et al., 2002, 2008; Lund et al., 2015a,b; Hessner et al., 2019) in Section 3.1.

**Changes in manuscript:** We edited the first paragraph of section 3.1 (lines 98 to 100) to include references discussing performances and limitations of WaMoS-II:

*"Performance of WaMoS-II and its limitations are discussed in Hessner et al. (2002, 2008, 2019); Lund et al. (2015a,b). A summary of the range and accuracy of measured parameters are reported in Appendix A."*

**Specific comments**

***R#1: Section 3.1, line 97: more details should be added on the type of wind sensor, its position on the vessel, the height measurement, the calibration procedure.***

**AC:** Details on the wind sensors are extensively discussed in Schmale et al. (2019); Thurnherr et al. (2020); Landwehr et al. (2020a). A brief summary describing the wind sensor has been added in Section 3.1 for completeness.

**Changes in manuscript:** We edited the text at the end of section 3.1 (lines 105 to 111) to add a few details about wind measurements:

*"... Further, water depth from the echo-sounder, ship's positions, speed, and course from a Global Positioning System (GPS) receiver and true wind velocity and direction from two two-dimensional sonic anemometers operating as part of an automated weather station (AWS) and mounted at 31.5 m above mean sea level (see Schmale et al., 2019; Landwehr et al., 2020a; Thurnherr et al., 2020) are fed into the system. Wind measurements were acquired at a rate of 1Hz, averaged over 175 s and converted from the measurement height to a neutral 10-metre wind speed ($U_{10}$) by assuming a logarithmic profile (see Holthuijsen, 2007) before being passed on to the WaMoS-II."*

***R#1: Line 120: please give more details on how the shadowing effects and tilting effects are removed. Is it a correction of a filter based on data quality control? How many data sets are eliminated by this procedure?***

**AC:** To minimize imaging effects like shadowing and tilt modulation, the WaMoS-II wave analysis was carried out in areas within a limited range (500-1500 m). This approach assumes that these effects are homogeneous and can be compensated by a single Modulation Transfer Function (MTF) as described in Nieto Borge et al. (2004). Note that this is a standard method in WaMoS-II and further details are not disclosed by the manufacturer and thus excluded. A brief discussion on mitigation of tilting and shadowing effects with the MTF (Nieto Borge et al., 2004) is added in Section 3.2.

The internal real time WaMoS-II quality control (iQC) protocol is independent of MTF. The iQC is based on various different individual tests (see Hessner et al., 2019), which evaluate conditions required for reliable radar-based wave measurements (e.g. sufficient sea clutter information or stable ship motion conditions). Most of the data that were labeled as unreliable were recorded in ice-covered waters with no significant surface waves present or the radar not operating in required short pulse mode. Therefore, data in the marginal ice zone has not been included in the data set. This has been remarked in the revised version of the manuscript.

**Changes in manuscript:** Few details about the removal of tilting and shadowing effects are provided in section 3.2 at lines 138 to 145:

*"... Radar imaging effects like tilt modulation, which refers to changes of the effective incidence angle along the long wave slope, and shadowing, which is caused by the highest waves in the image, contribute to an inaccurate form of the resulting spectral density function, shifting energy towards high wavenumbers (Nieto Borge et al., 2004). These effects depend on the view geometry (height and range of the antenna). Consequently, tilting and shadowing can be assumed to be homogeneous in the relatively small sub-areas used for post processing and can be minimised with a single modulational transfer function (MTF, Nieto Borge et al., 2004). As the imaging effects depend on the wavelength, the MTF is a function of the wavenumber that corrects the spectral density at each mode. An ensemble average over all sub-areas is computed to derive the final wave spectrum $E(k_x, k_y)$ from the input 64 images."*

We have also modified the last paragraph of section 3.2 (lines 165 to 171) to include details of the quality control process (see previous comment). The new text reads as

*"Rain, snow and sea ice produce an excess signal backscatter, which results in low quality images and consequently inaccurate post processing products. WaMoS-II automatically assesses*

*the reliability of images through an internal quality control protocol (see Hessner et al., 2019), which evaluate backscatter intensity, number of sea clutters and stability of ship motion among other parameters (we remark that tilting and shadowing effects are compensated independently using the MTF and do not contribute to quality control). Images that are deemed of low quality are excluded. The majority of low quality images were acquired in the marginal ice zone (i.e. south of the $60^{th}$ parallel) during Leg 2. As a consequence, observations of waves-in-ice are not available in the present database."*

***R#1: Line 124 and following: the method for rescaling the wave spectrum deserves more details. Indeed, I could not find details on this rescaling in the Young et al. (1985) publication. Furthermore, other publications on WAMOS, like the one of Nieto Borge et al. (2004) mention that this type of rescaling may not be fully appropriate, as the Transfer Function between image intensity and wave heights depends on the wave number of the ocean waves. Could you comment on that in the manuscript?***

**AC:** We thank the reviewer for this comment. The rescaling that we used is the standard method implemented in the WaMoS-II software. Available details can be found in Nieto Borge et al. (1999, 2004). These details are already summarised in the original manuscript (lines 124 and following). Note that additional, more detailed, information about the rescaling is not disclosed by the manufacturer and thus cannot be presented.

The reviewer is right when stating that no information is provided by Young et al. (1985) in this regard. Reference to that paper was included by mistake and the correct citations have been added to the revised version of the manuscript.

The capabilities of the rescaling technique are discussed and demonstrated in Nieto Borge et al. (1999) and Hessner et al. (2002); these two references have been added to the revised version of the manuscript and briefly commented on. We do not find any discussion on the appropriateness of rescaling techniques in Nieto Borge et al. (2004) as mentioned by the reviewer. If we misunderstood the comment, we would appreciate receiving more details in order to better address this concern. Nevertheless, our understanding of the discussion in Nieto Borge et al. (2004) is that the Modulation Transfer Function (MTF) compensates for the nonlinearities related to imaging effects such as tilt modulation and shadowing. As these depend on the view geometry (antenna height and range), WaMoS-II limits the analysis range to an area where the imaging effects are assumed to be homogeneous, allowing application of the MTF method. As the imaging effects also depend on the wavelength, the MTF is a function of wavenumber. While the MTF allows extrapolating the wave spectrum from the image spectrum, the resulting spectral density still requires a calibration to return the correct energy content (the MTF returns the relative energy distribution and not the absolute one). In order to get the absolute values, the spectrum needs to be rescaled, so that the individual frequency and direction bins are related to wave energy. To do so, WaMoS-II uses the rescaling techniques discussed at lines 124 and following. The text has been reworded to clarify the overall process.

**Changes in manuscript:** We edited the description of the post processing of the radar images in section 3.2. This includes the re-scaling process (lines 131 to 138):

*"... The latter* [image spectrum]*, however, does not coincide with the wave energy spectrum, because it represents the intensity of the radar backscatter rather than the amplitude of the water surface elevation (Nieto Borge et al., 1999; Hessner et al., 2002). Therefore, its zero-th order*

*moment ($m_0$) represents a signal-to-noise-ratio (SNR) instead of the significant wave height, i.e. a measure of average wave height that is defined as $H_s = 4\sqrt{m_0}$. Consequently, the image spectrum requires a re-scaling to convert SNR into the corresponding wave height. This is achieved with the linear regression equation (see Nieto Borge et al., 1999, 2004)*

$$H_s = A + B\sqrt{SNR},\tag{1}$$

*where A and B are empirical constants that have to be calibrated following installation. Re-scaling $m_0$ enables correction of energy at each spectral mode and derivation of the wave energy spectrum $E_r(k_x, k_y)$. ..."*

and removal of tilting and shadowing effects (lines 138 to 144):

*"... Radar imaging effects like tilt modulation, which refers to changes of the effective incidence angle along the long wave slope, and shadowing, which is caused by the highest waves in the image, contribute to an inaccurate form of the resulting spectral density function, shifting energy towards high wavenumbers (Nieto Borge et al., 2004). These effects depend on the view geometry (height and range of the antenna). Consequently, tilting and shadowing can be assumed to be homogeneous in the relatively small sub-areas used for post processing and can be minimised with a single modulational transfer function (MTF, Nieto Borge et al., 2004). As the imaging effects depend on the wavelength, the MTF is a function of the wavenumber that corrects the spectral density at each mode. ..."*

**R#1: Line 132-133: please give details or references on how the partitions were estimated (method of partitioning, external data used in the partitioning if any - like wind speed and wind direction,...)**

**AC:** The partitioning of the wave spectrum is performed using the "path of steepest ascent" technique proposed by Hanson and Phillips (2001), which is a specific implementation of the inverse catchment scheme introduced by Hasselmann et al. (1996). Different wave systems (i.e. spectral peaks or subcatchments) are determined by associating each spectral grid point to the neighbour with the highest energy level. Grid points corresponding to the same local peak are clustered, and each of these clusters defines a partition (watershed algorithm). The spectral peak that satisfies the condition:

$$1.2\frac{U}{c_p}cos(\theta - \psi) > 1,\tag{2}$$

where $U$ is the wind speed, $c_p$ is the phase velocity, $\theta$ is the wave direction and $\psi$ is the wind direction, is associated with the wind sea. All other systems are swell and are ranked based on their energy contents as primary, secondary and tertiary swell. Details on partitioning and related references have been added to the revised version of the manuscript in Section 3.2.

**Changes in manuscript:** We added the following sentences to section 3.2 at lines 155 to 161:

*"The partitioning of the wave spectrum is performed using the "path of steepest ascent" technique (Hanson and Phillips, 2001), which is a specific implementation of the inverse catchment scheme introduced by Hasselmann et al. (1996). The spectral peak that satisfies the condition*

$$1.2\frac{U}{c_p}\cos(\theta - \psi) > 1,\tag{3}$$

*where U is the wind speed, $c_p$ is the phase velocity, θ is the wave direction and ψ is the wind direction, is assumed to be associated to the wind sea. All other systems are swell and are ranked based on their energy contents as primary, secondary and tertiary swell."*

**R#1: Section 3.4: I am surprised that only ship data are used to build reference values of significant wave height Hs. You do not have any possible comparison with buoy data when the ship was in coastal regions? Using ship IMU data as reference to obtain Hs does not seem so trivial as shown for example by Nielsen and Dietz (2020) (see e.g. "Estimation of sea-state parameters by the wave buoy analogy with comparisons spectral wave models≫, Ocean Engineering 2020). In the ship to wave spectral transformation, do you take into account the possible non-linearities of the ship response, the effects of ship speed, of direction of waves with respect to the ship heading,....? More details should be added in this section. On the other hand, I must admit that the a posterior validation using satellite significant wave heights, as presented in Fig.7, is convincing.**

**AC:** That is correct, we could not perform a calibration based on buoy data because there were no co-located buoy measurements during ACE. Therefore, the calibration had to rely on the sea state retrieved from ship motion data. The underlying principle for sea state reconstruction is that the ship is a rigid body with six degrees of freedom (three translations: heave, surge, and sway; and three rotations: pitch, roll, and yaw) and moves in response to the incident wave field and restoring forces as a function of its mass, geometry, loading conditions and forward speed among other parameters (Newman, 2018). The relation linking ship motion to the energy spectrum of the incident wave field is evaluated by the response amplitude operator ($R(f)$, see Newman, 2018), i.e. a ship-specific function that translates the motion spectrum ($S_{ship}(f)$) into the incident wave spectrum ($S_{wave}(f)$): $R(f)^{-2} = S_{wave}(f)/S_{ship}(f)$.

Motion spectra were evaluated by applying a Fourier Transformation over 5 minute long time series of heave motion. $R(f)$ was calculated by solving the equation of motion with a model based on the boundary element method (Babarit and Delhommeau, 2015), taking into account the ship's heading, forward speed and loading conditions; the model is based on a linear approach and thus nonlinearities were excluded. The significant wave height was validated against freely available satellite altimeter data (Ribal and Young, 2019) of significant wave height for the entire voyage. The text describing the sea state reconstruction from ship motion has been updated (see Section 3.4). Furthermore, we have added an appendix (Appendix B) in the revised manuscript, where we discuss the accuracy of the reconstructed sea state against satellite altimeter observations. The appendix includes Figure B1, which shows the scatter plot of significant wave height from satellite altimeter versus significant wave height reconstructed from the ship motion. Due to the coarse resolution of satellite data, average values are computed for clusters with spatial resolution of $0.5° \times 0.5°$ and temporal resolution of 3 hours. There is a good agreement overall. The root-mean squared error ($RMSE$) is $\approx 0.4\,\mathrm{m}$, the correlation coefficient ($R$) is $\approx 0.94$, and the scatter index ($SI$) is $\approx 0.17$. Note that similar error metrics have been obtained by comparing the reconstructed sea state against parameters from European Centre for Medium-Range Weather Forecasts (ECMWF–`https://www.ecmwf.int/en/forecasts/datasets/reanalysis-datasets/era5`) ERA-5 reanalysis.

**Changes in manuscript:** Section 3.4 (lines 201 to 204) has been edited to improve clarity, but no significant changes have been made. Appendix B has been added at lines 389 to 395 to briefly discuss the validity of the reconstructed sea state from ship motion. A figure showing

a comparison of reconstructed $H_s$ versus satellite observations has been added. The text and figure are as follow:

"*Significant wave heights reconstructed from ship motion data were validated against freely available satellite observations Ribal and Young (2019) for the entire ACE voyage (see scatter plot in Figure B1). Due to the coarse resolution of satellite data, average values are computed for clusters with spatial resolution of $0.5° \times 0.5°$ and temporal resolution of $3\,h$. Overall, there is a good agreement between reconstructed and observed sea state. The root-mean squared error (RMSE) is $0.4\,m$, the correlation coefficient (R) is 0.94, and the scatter index (SI) is 0.17. Similar error metrics is obtained by comparing the reconstructed sea state against parameters from the European Centre for Medium-Range Weather Forecasts (ECMWF–`https://www.ecmwf.int/en/forecasts/datasets/reanalysis-datasets/era5`) ERA-5 database.*"*

[Figure]

Figure 1: This is Figure B1 in the revised manuscript—Satellite observations versus significant wave height reconstructed from ship motion.

***R#1: Section 4.1 comments about the statistics on current: You have omitted to mention that the current from satellite altimeters are not surface currents but geostrophic currents.***

**AC:** We thank the reviewer for spotting this error. We used current data from COPERNICUS-GLOBCURRENT - `https://marine.copernicus.eu`, which combines the total velocity field based on satellite geostrophic surface currents with modelled Ekman currents, which includes wind stress forcing obtained from atmospheric system and drifters data. Information on COPERNICUS-GLOBCURRENT has been added in Section 4.1.

**Changes in manuscript:** The first paragraph of section 4.1, lines 218 to 221, has been edited as follows:

"*... Data of wind speed and wave height are from all satellite missions mounting altimeter sensors that are available from 1985 to 2019 (Ribal and Young, 2019). Data of current speed*

*are from the COPERNICUS-GLOBCURRENT database—$https://marine.copernicus.eu$—
that combines the velocity field of geostrophic surface currents from satellite sensors recorded
from 1993 to 2019 (Rio et al., 2014) and modelled Ekman currents, which include components
from wind stress forcing obtained from atmospheric system and drifters data."*

**R#1: Section 4.2 lines 204-207: please , indicate how the raw wind measurements
were converted into ten meter-height winds (U10), and what is the duration of
integration of the raw data.**

**AC:** Wind speed and directions were recorded at the sampling rate of 1Hz and averaged over
a period of 175 s period and converted from the measurement height to a 10-metre above sea
level wind speed ($U_{10}$) by assuming a logarithmic profile (see Landwehr et al., 2020a), before
being passed on to WaMoS-II. Atmospheric boundary layer instability is not considered and
thus $U_{10}$ represents the neutral wind speed. Details have been added in Section 3.1.

**Changes in manuscript:** The last paragraph of section 3.1, lines 109 to 111, has been edited
as follows:

*"... Wind measurements were acquired at a rate of 1Hz, averaged over 175 s and converted
from the measurement height to a neutral 10-metre wind speed ($U_{10}$) by assuming a logarithmic
profile (see Holthuijsen, 2007) before being passed on to the WaMoS-II."*

**R#1: Line 223-224: it is strange that the only references that you give to mention
the oceanic directional distribution of waves come from wave tank measurements.
Could you add some references on field measurements?**

**AC:** References to field measurements by Mitsuyasu et al. (1975); Donelan et al. (1985); Young
and Verhagen (1996); Young et al. (2020) have been added.

**Changes in manuscript:** Lines 276 to 279, have been edited as follows:

*"An intrinsic feature of oceanic sea states is the directional distribution of the spectral
density function (Mitsuyasu et al., 1975; Donelan et al., 1985; Young and Verhagen, 1996;
Toffoli et al., 2017; Fadaeiazar et al., 2020; Young et al., 2020), which is summarised in the
form of a mean directional spreading (i.e. the circular standard deviation of the directional
wave energy spectrum). ..."*

**R#1: Line 240: here again , mention that the current measurements from WAMOS-
II and the climatological currents estimated from altimeter data do not represent
exactly the same geophysical quantity.**

**AC:** The reviewer is right. Even though we used a combination of geostrophic surface currents
and modelled Ekman currents, WaMoS-II still detects additional components such as inertial
oscillations (Treguier and Klein, 1994), which are not represented in the benchmark data set.
As inertial oscillations are particularly strong in the Southern Ocean, they represent a notable
source of uncertainty. Furthermore, Ekman components remain uncertain in the Southern
Ocean due to inaccuracies in estimating wind stress from the atmospheric system, adding
more inconsistencies between benchmark and our observations. In the revised version of the
manuscript we have commented on the differences between our observations and the benchmark
data in Section 4.2.

**Changes in manuscript:** We edited discussion of current speed during ACE in section 4.2 (lines 270 to 275) as follows:

*"... We remark that P50 and P90 include the contribution of geostrophic surface currents and wind stresses. However, additional components such as inertial oscillations (Treguier and Klein 1994) are not taken into account due to the course resolution of satellite observations and Ekman components. To some extent, the absence of inertial oscillations in climate statistics substantiate the significant current speeds recorded by the WaMoS-II. Further, Ekman components remain uncertain in the Southern Ocean due to inaccuracies in estimating wind stress from the atmospheric system, adding inconsistencies to benchmark statistics."*

**R#1: Lines 270-273: you could mention that on these examples, SAR does not detect the wind sea, in opposite to WAMOS-II data.**

**AC:** We thank the reviewer for this comment. We have added an additional statement to stress that SAR does not detect wind sea.

**Changes in manuscript:** Text at lines 318 to 325 in section 5.2 has been changed with teh following:

*"Examples of collocated wave spectra from WaMoS-II and SAR are presented in Fig. 9; mean wind direction is also reported. We remark that SAR detects wavelength longer than 115 m (approximately, wave periods exceeding 8 s or frequencies below 0.1 Hz) and represented swell systems primarily. WaMoS-II, on the contrary, captures the full spectrum, including the short wavelengths of the wind sea. Within the operational range of SAR ($f < 0.1\,Hz$ in the figure), the spectral shape from both sensors agrees well, especially for the portion around the primary (most energetic) swell. Notable discrepancies, however, are evident for less energetic secondary peaks, for which the relative uncertainty grows. High frequency components ($f > 0.1\,Hz$) are not resolved in SAR, but appear in the WaMoS-II spectra. Note that the misalignment of high frequency components with the wind direction in the upper two panels is due to recent wind change."*

**Technical corrections**

**R#1: Figure 5: you could mention in the legend that the circles in dashed light lines (hardly visible) are plotted every 15° in latitude**

**AC:** We have edited Figure 5 to make the circles slightly thicker and added details about their meaning in the caption.

**R#1: Line 201: "pattern"(instead of "patter")**

**AC:** This typo has been corrected.

**R#1: Figure 8 i) the marks for the scales are not visible (circles in wave number or frequency) ii) Also could you add the wind direction on these polar plots?**

**AC:** The marks for the scales in Figure 8 (Figure 9 in the revised manuscript) has been made thicker. Wind directions have been added.

**Response to Anonymous Reviewer #2**

*R#2: Line 106-107: Do you have an estimate of what proportion of the data (if any?) is not processed due to the ocean being too smooth? Is this much of an issue in the Southern Ocean?*

**AC:** Although it is unusual for the Southern Ocean, there were conditions of very low wind speed, resulting in a smooth surface. In more general sense, WaMoS-II cannot detect the ocean surface accurately if wind speed is lower than $3\,m/s$. Low wind speed affected about 9% of the whole observation. We have commented on the low wind speed issue in the revised version of the manuscript in Section 3.2.

**Changes in manuscript:** This sentence has been added to section 3.2 (lines 117 to 118):

*"... A small portion of the observations during ACE (approximately 9%) were taken during low wind speed and hence removed from the data set."*

*R#2: Line 128: "modes" = "mode"?*

**AC:** This typo has been corrected (line 138).

*R#2: Line 180: Is there a standard deviation or other measure you could include to show the variability of these variables over the 20-30 year time period? (This might be useful to include in Fig 6 to put your instantaneous observations in context with the inter annual variability?)*

**AC:** We thank the reviewer for this comment. We have added the interquartile range (IQR) associated to P50. To make the IQR more visible, we have split the original Figure 6 into two figures in the revised manuscript. Figure 6 shows variables for which climate statistics are available; Figure 7 shows all other variables.

**Changes in manuscript:** The new Figure 6 in the revised manuscript is reported in Figure 2 in this document.

*R#2: Figure 5: perhaps add a contour showing the sea ice edge to the figures that show wave height to highlight the attenuation you mention in line 195? (Figure 5c and/or Figure 5d).*

**AC:** We updated Figure 5 by adding the contour line of the sea ice edge as located at 10% ice concentration.

**Changes in manuscript:** The new Figure 5 in the revised manuscript is reported in Figure 3 in this document.

*R#2: Line 202: "patter" = "patterns"*

**AC:** This typo has been corrected (line 239).

*R#2: Line 202: maybe add either the polar front or the legs to the figures showing surface currents to highlight the region you are describing here? It would be useful to reference Figure 5f here.*

[Figure]

Figure 2: This is Figure 6 in the revised manuscript—Time series of sea state variables in Leg 1 (blue), Leg 2 (green), and Leg 3 (red): (a) wind speed from the automated weather station; (b) significant wave height; and (c) current speed. For each variable, the dashed line and shading represent $50^{th}$ percentile and its interquartile range IQR, respectively, based on climate statistics from satellite observations; the solid line indicates the $90^{th}$ percentile.

**AC:** Voyage's route has been added in Figure 5 (see Figure 3 of this document).

***R#2: Figure 7: c and/or d - add the sea ice edge (or include the shading from Fig 1) so we can visualize where the sea ice is complicating estimations.***

**AC:** Sea ice has been added to panels c and d of this figure. Note that Figure 7 in the original manuscript is Figure 8 in the revised version.

**Changes in manuscript:** The new Figure 8 in the revised manuscript is reported in Figure 4 in this document.

***R#2: Line 283: Is there any other literature on WaMoS-II surface current observations that provides any assessment of the quality of the current observations? I.e. are you able to say anything about the quality of the WaMoS-II vs altimeter current measurements? How different are the current estimates - maybe it isn't appropriate to directly compare these quantities?***

**AC:** We thank the reviewer for this comment. The accuracy of the ocean current measurements is discussed in Lund et al. (2015a,b); Hessner et al. (2019). A brief comment about accuracy with reference to the aforementioned papers have been added (Section 3.1).

[Figure]

Figure 3: This is Figure 5 in the revised manuscript—Wind speed ($U_{10}$), significant wave height ($H_s$), and surface current speed ($u$) climatology in Austral summer: (a) $50^{th}$ percentile (median) wind speed, (b) $90^{th}$ percentile wind speed, (c) $50^{th}$ percentile (median) significant wave height, (d) $90^{th}$ percentile significant wave height, (e) $50^{th}$ percentile (median) surface current speed, (f) $90^{th}$ percentile surface current speed. Latitudes are shown every $15°$ (from $15°S$ to $90°S$) by thin lines; the route of the ACE voyage is reported as a black solid line); and the sea ice edge, defined by the 10% sea ice concentration, is shown as a grey solid line.

[Figure]

Figure 4: This is Figure 8 in the revised manuscript—Wind from the automated weather station and significant wave height from WaMoS-II versus satellite observations: scatter diagrams (panels a and b); and geographical distribution of biases (panels c and d). Average sea ice concentration during the expedition is overlaid in panels c and d.

Regarding satellite observations, we used current data from COPERNICUS-GLOBCURRENT - `https://marine.copernicus.eu`, which provides the total velocity field based on satellite geostrophic surface currents and modelled Ekman currents, which take into account wind stress forcing obtained from atmospheric system and drifters data. In principle, this product is consistent with current measurements from WaMoS-II. Nevertheless, WaMoS-II also detects inertial oscillations (Treguier and Klein, 1994), which are not detected by satellite observations and Ekman components due to their coarse resolution but can be particularly intense in the Southern Ocean. These components, however, are not captured by WaMoS-II and hence represent a source of inconsistency. A remark in this regard has been added to the manuscript in Sections 4.2 and 5.3.

**Changes in manuscript:** We have added the following text to section 3.1 (lines 98 to 100):

*"Performance of WaMoS-II and its limitations are discussed in Hessner et al. (2002, 2008, 2019); Lund et al. (2015a,b). A summary of the range and accuracy of measured parameters are reported in Appendix A."*

We edited discussion of current speed during ACE in section 4.2 (lines 270 to 275) as follows:

*"... We remark that P50 and P90 include the contribution of geostrophic surface currents*

*and wind stresses. However, additional components such as inertial oscillations (Treguier and Klein 1994) are not taken into account due to the course resolution of satellite observations and Ekman components. To some extent, the absence of inertial oscillations in climate statistics substantiate the significant current speeds recorded by the WaMoS-II. Further, Ekman components remain uncertain in the Southern Ocean due to inaccuracies in estimating wind stress from the atmospheric system, adding inconsistencies to benchmark statistics."*

and section 5.3 (lines 342 to 344):

*"... The reported differences are linked to inconsistencies between WaMoS-II and benchmark data due to inertial oscillations (Treguier and Klein, 1994), which are not detected by satellite observations, and inaccuracy of wind stresses in the Ekman components."*

**Response to Anonymous Reviewer #3**

*R#3: I would like a brief comment by the authors on the spectral frequency resolution and parameters variability that originates from the 160-second long duration of Wamos-II acquisition (if I correctly understood). I mean, with 160-s long records the spectral resolution over frequency is very low (large delta f). And given the rotation speed of the antenna I guess also the maximum resolved frequency may be very low. How does this affect the spectral representation? In addition, with Tm > 8 s (also 13 s) waves every sample includes less than 20 waves. So, I suspect the estimate of the spectrum and wave parameters (even including the 600×1200 m2 area) might be pretty unstable.*

**AC:** The X-band radar provides spatio-temporal information by recording 64 images of the surrounding ocean surface over a period of 175 s (we erroneously indicated 160 s in the original submission). This corresponds to one image for each full turn of the antenna. A single wavenumber spectrum is computed from all images recorded within this period of time. Specifically, the spectral density is estimated by applying a three dimensional Discrete Fourier Transform to three sub-areas of dimensions 600 m × 1200 m for each image/surface. The final spectrum is computed as an ensemble average over the 175 s and all sub-areas. Therefore, the spectral resolution is dictated by the spatial resolution of the sub-areas and not the temporal duration of the sampling. Considering the resolution of the image (5 m) and the minimum dimension of the sub-area (600 m), WaMoS-II can detect wavelengths between 10 m and 600 m which correspond to wave periods from 3 s to ≈16 s. We have added a comment on the spectral resolution in the revised version of the manuscript (Section 3.2).

**Changes in manuscript:** We have added the following text to section 3.2 (lines 148 to 151):

*"... The resolution of the wave energy spectrum is dictated by the size of the sub-areas, which are used to derive the wavenumber counterpart in the first instance, and not by the temporal window. Considering the resolution of the image (12 m) and the minimum dimension of the sub-area (600 m), WaMoS-II can detect wavelengths between 15 m and 600 m, which correspond to wave periods from 3 s to ≈ 16 s."*

*R#3: Wind data are measured by an on board meteorological station, but in Figure 7 the measured wind U10 is labeled as Wamos-II. Please may you check consistency?*

**AC:** We thank the reviewer for spotting this error. Label of wind speed has been corrected in the revised version (note that Figure 7 in the original submission is now Figure 8 in the revised version).

**Changes in manuscript:** We have updated the figure (now Figure 8). See the new figure below for convenience.

[Figure]

Figure 5: This is Figure 8 in the revised manuscript—Wind from the automated weather station and significant wave height from WaMoS-II versus satellite observations: scatter diagrams (panels a and b); and geographical distribution of biases (panels c and d). Average sea ice concentration during the expedition is overlaid in panels c and d.

**R#3: In Figure 8, axis labels (units and variables) are missing.**

**AC:** Figure 8, now Figure 9 in the revised manuscript, shows polar plots of the directional wave spectra. There are no axis labels for this type of plot normally. The colorbar represent the normalised energy density and thus it is non-dimensional. We have updated the caption to make sure the all details of the figure are clear.

**Changes in manuscript:** Labels of Figure 9 (former Figure 8) have not been changed. Figure's caption has been updated with the following text:

[revised manuscript text omitted]